# Bioinspired thermadapt shape-memory polymer with light-induced reversible fluorescence for rewritable 2D/3D-encoding information carriers

Jinhui Huang[1,2], Yue Jiang[1,2], Qiuyu Chen[1,2], Hui Xie [1,2] & Shaobing Zhou [1,2]

Fluorescent materials have attracted widespread attention for information encryption owing to their stimuli-responsive color-shifting. However, the 2D encoding of fluorescent images poses a risk of information leakage. Herein, inspired by the mimic octopus capable of camouflage by changing colors and shapes, we develop a thermadapt shape-memory fluorescent film (TSFF) for integrating 2D/3D encoding in one system. The TSFF is based on anthracene group with reversible photo-cross-linking and poly (ethylene-co-vinyl acetate) network with thermadapt shape-memory properties. The reversible photo-cross-linking of anthracene is accompanied by repeatable fluorescence-shifting and enables rewritable 2D encoding. Meanwhile, the thermadapt shape-memory properties not only enables the reconfiguration of the permanent shape for creating and erasing 3D patterns, i.e., rewritable 3D information, but also facilitates recoverable shape programming for 3D encoding. This rewritable 2D/3D encoding strategy can enhance information security because only designated inspectors can decode the information by providing sequential heating for shape recovery and UV exposure. Overall, TSFF capable of rewritable 2D/3D encoding will inspire the design of smart materials for high-security information carriers.

With the rapid development of the information age, global data storage will exceed 200 zettabytes by 2025[1,2]. Correspondingly, problems can be caused by counterfeit information and private important information leakages[2]. For example, counterfeit banknotes, tax stamps, certificates, and trademarks in the pharmaceutical and food industries have led to serious social problems. Researchers have been working on various emerging anti-counterfeiting and information encryption materials and strategies for a long time, including holography approaches, multidimensional security codes, and smart fluorescent materials[3–7]. Recently, smart fluorescent materials have attracted widespread attention for information storage and encryption owing to their color-changing in response to the right external stimuli, such as light, heat, chemicals, and/or electricity[8–11]. However, the information (e.g., images) encrypted by the fluorescence is two-dimensional (2D). As such, it can be easily decrypted, for example, by irradiating with a commercial ultraviolet (UV) light. Thus, the 2D-encoding strategy poses a risk of information leakage.

The imperfection of the single 2D-encoding module has pushed researchers to integrate it with three-dimensional (3D)-encoding into one system. The key to 3D-encoding is employing shape-programmable smart materials; among these, shape-memory polymers (SMPs) are popular[12–15]. SMPs facilitate programming via

[1]Institute of Biomedical Engineering, College of Medicine, Southwest Jiaotong University, 610031 Chengdu, China. [2]Key Laboratory of Advanced Technologies of Materials Ministry of Education, School of Materials Science and Engineering, Southwest Jiaotong University, 610031 Chengdu, China. e-mail: huixie@swjtu.edu.cn; shaobingzhou@swjtu.edu.cn

deforming and fixing into a temporary new 3D shape and recovering to the original shape in response to the right stimulus, for example, heat[16]; this feature enables broad applications, such as intelligent robots, flexible electronics, biomedical devices, and information camouflaging[17–19]. An SMP can be curled or folded into 3D geometries to hide the 2D fluorescent pattern; in this way, the true information can only be decrypted once the SMP recovers to its unfolded state. As reported by Wu et al., a shape-memory hydrogel containing blue fluorophores was engineered as a dual-encrypted information medium, in which a 2D fluorescent pattern was temporarily encased by folding the flat hydrogel for shape programming[20]. This study provided inspiration for integrating 2D and 3D-encoding. Similar work also reported by Wu et al. who developed a boroxine-based material for dual-encryption[21]; the 2D information was protected by subsequent 3D deformation. Despite the above successes, hydrogels are susceptible to physical damages from external stress, resulting in undesired identification of the 3D-encased 2D information and thereby weakening the long-term security; in addition, hydrogels inevitably become dehydrated during use[22–24]. Although bulk polymers may be affordable for long-term use, the major concern for the current works is that the shape-memory effect (SME) is merely used for information encasement in new shapes; it does not benefit the creation of 3D patterns carrying practical information towards high-security at the same time. Thus, it is expected that practical 3D information can be encrypted by SME and integrated with a 2D fluorescent pattern, thereby realizing the true integration of 2D/3D-encoding and improving security.

Usually, 3D information can be represented as a macro-, micro-, or nano-pattern printed on an information carrier. The commercial technique for creating 3D information uses certain template and direct molding or etching[25,26]. Meanwhile, 3D printing has also been used to create high-resolution 3D patterns[27]. However, these 3D patterns cannot be reprogrammed, making the 3D information not erasable or rewritable; this is not beneficial to a sustainable material platform. Based on the advantages of the shape-shifting feature of SMPs, researchers have also used templates when deforming SMPs and constructed certain 3D patterns[28–30]. However, for information carriers, one contradiction is that a 3D pattern created in this way will be wiped when using the SME to encase it; thus, it cannot be encrypted. Therefore, SMPs capable of reconfiguring their permanent shape are in high demand; in this context, such an SMP has two functions, the shape reconfiguration facilitates the creation of rewritable 3D information, and the SME can be used for encasing the information subsequently. This is the focus of our proposal for 3D-encoding. Recently, thermadapt SMPs containing dynamic bonds have become increasingly popular because they can reconfigure their permanent shapes through the reversible dissociation and reassembly of their dynamic bonds[31,32]. Both dynamic chemical covalent bonds and non-covalent bonds (such as ester, imine, disulfide, urethane, and hydrogen bonds) are available for developing thermadapt SMPs[33,34]. From the perspective of 3D-encoding, a thermadapt SMP not only enables the creation of a 3D pattern (information input) by activating dynamic bonds, but also encases the 3D information using the SME (3D-encoding). This is a very attractive approach for integrating 2D/3D-encoding. However, the incorporation of dynamic bonds into an SMP always requires a complicated chemical synthesis, making it difficult to prepare anticounterfeiting materials for practical use.

Herein, inspired by the mimic octopus, one of nature's camouflage masters capable of changing its color and shape to adapt to the surrounding environment or escape from predators (Supplementary Fig. 1), we designed and fabricated a type of thermadapt shape-memory fluorescent film (TSFF) using commercial poly (ethylene-co-vinyl acetate) (EVA) and anthracene for a rewritable 2D/3D-encoding information carrier (Fig. 1). Anthracene has a reversible photo-cross-linking nature upon irradiation with UV light with different wavenumbers (365 and 254 nm), in addition, a sharp contrast exists in the fluorescence before and after photo-cross-linking[35,36]. This enables the creation of a 2D fluorescent pattern and its erasure, i.e., rewritable 2D information. Meanwhile, the ester bonds in the cross-linked EVA (cEVA) can be dynamic[37], enabling the creation and erasure of 3D patterns (rewritable 3D information); in addition, the cEVA holds an excellent thermo-responsive SME, ensuring the shielding of the 3D information. In this way, 2D/3D-encoding can be achieved. The TSFF is also weldable and reprocessable, making it a multifunctional information carrier.

## Results

### Design and fabrication of thermadapt shape-memory fluorescent film (TSFF)

In this work, the TSFF was prepared from EVA and acryloylated anthracene-containing polycaprolactone (AC-PCL-AN) by using dicumyl peroxide as the cross-linker. The molding process was similar to those in our previous works[38,39]. In particular, 1,5,7-triazabicyclo [4.4.0] dec-5-ene (TBD) was also added during preparation to act as a transesterification catalyst for activating the dynamic nature of the cEVA. In this way, the reversible fluorescence for rewritable 2D encoding and thermadapt shape-memory properties for rewritable 3D encoding were expected to be realized by the PCL-AN and cEVA components, respectively.

First, the AC-PCL-AN was synthesized using two steps: a ring-opening polymerization of the ε-Caprolactone by 9-Anthracenemethanol and the introduction of C = C bond by acryloyl chloride (Supplementary Fig. 2). The successful preparation of the AC-PCL-AN was confirmed by the results from proton nuclear magnetic resonance ($^1$H NMR), where the characteristic peaks of the PCL segment, anthracene group, and C = C bond could be identified (Supplementary Fig. 3). Then, the AC-PCL-AN and EVA were homogeneously mixed in toluene and dried, hot pressed at 100 °C, and eventually cross-linked at 170 °C to obtain the TSFF. The modification of the anthracene group on the PCL could potentially promote its interdiffusion and moveability in the cEVA; this was expected to provide a more uniform fluorescence distribution and higher efficiency of the light-induced reversible cross-linking relative to a method based on directly adding the anthracene-containing compound into the matrix. In particular, cEVA contains an abundance of ester groups (Supplementary Fig. 4) which could service as thermally induced dynamic cross-links to enable shape-reconfiguration of TSFF for creating 3D patterns.

Overall, the TSFF with light-reversible fluorescent moieties and thermally dynamic covalent bonds could satisfy the requirements for chemical structures for the integration of 2D encoding based on fluorescence-shifting and 3D encoding based on shape-shifting. The following sections are focused on the investigations of several key properties of the multifunctional TSFF for rewritable 2D/3D-encoding information carriers, in particular, the fluorescence-shifting and thermadapt shape-memory properties.

### Reversible fluorescence-shifting properties

Anthracene shows reversible photo-cross-linking when irradiated by alternating UV light (365 and 254 nm) (Fig. 2a); the change of fluorescence is distinct, laying the foundation for photo-patterning. As shown in Fig. 2b, as observed by a UV-Vis spectrometer, the absorbance of anthracene group in the TSFF gradually decreases with the increase of the irradiation time of the 365 nm UV light. As fluorescence was the core of the current work, we further examined the fluorescence change of the TSFF upon 365 nm UV irradiation. As shown in Fig. 2c, the fluorescence intensity recorded by the fluorescence spectroscopy decreases dramatically when the irradiation time with 365 nm UV light is increased. These results confirm the photo-cross-linking of the anthracene group, which can undergo [4 + 4] cycloaddition under 365 nm UV irradiation. Consequently, the conjugated structure

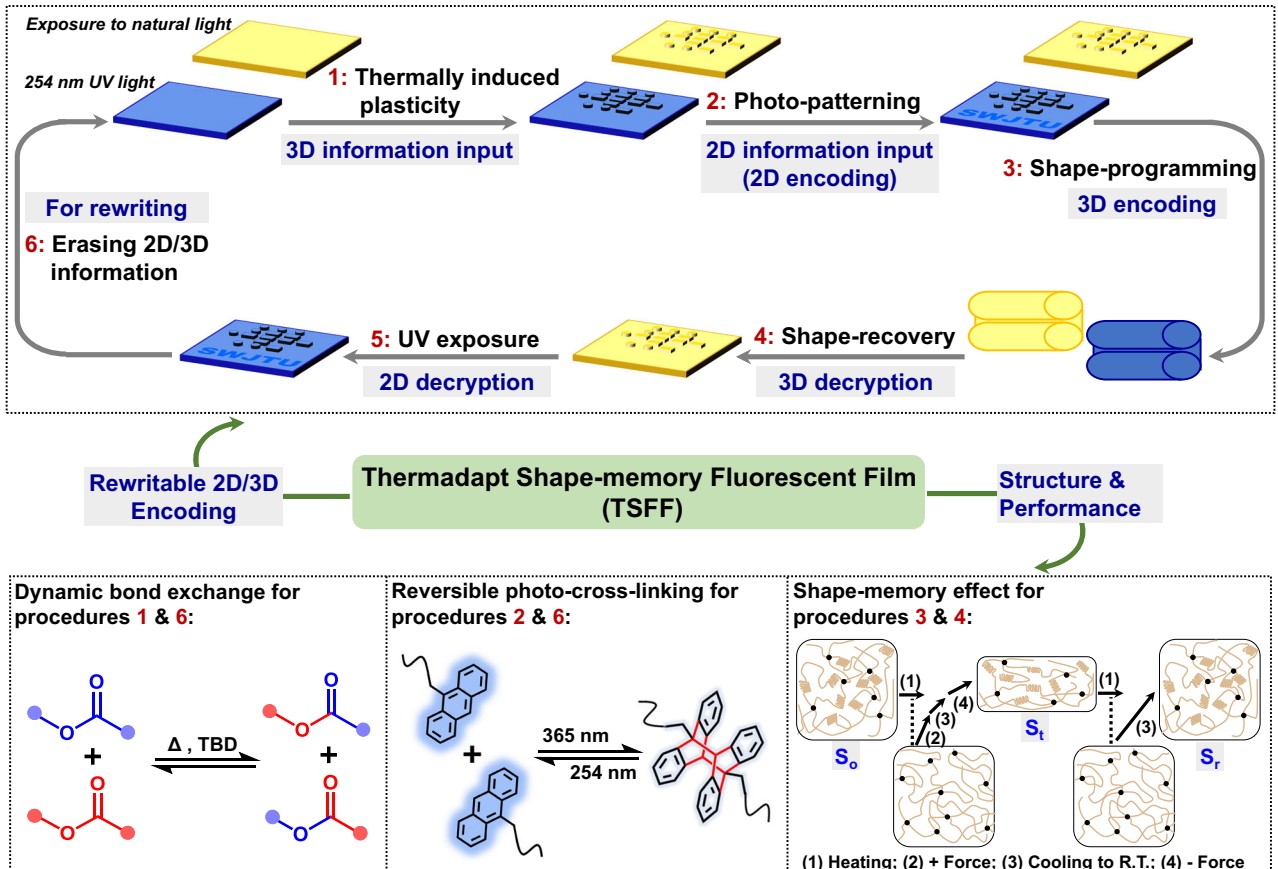

**Fig. 1 | Schematic illustration of the rewritable 2D/3D-encoding information carriers based on the multifunctional thermadapt shape-memory fluorescent film (TSFF).** The rewritable 2D/3D encoding includes six main procedures: (1) the input of 3D information by thermally induced plasticity of TSFF, the 3D pattern is visible, so the information is not encrypted; (2) the input of 2D information by photo-patterning, the 2D image is invisible under natural light but visible when exposure to UV light (254 nm), so 2D encoding is achieved; (3) 3D encoding by shape-programming, both the 2D and 3D information are invisible; (4) decoding of 3D information after shape-recovery; (5) decoding of 2D information by exposing to UV light (254 nm); (6) erasing the created 3D and 2D information by thermally induced plasticity and reversible photo-cross-linking, which enables rewriting information in successive cycles. The figures at the bottom display the chemical structures, reactions, and shape-memory procedures for rewritable 2D/3D encoding.

dominating the fluorescence emission should ultimately be destroyed and eventually lead to the decline of the fluorescence.

In accordance with the requirements for rewritable photo-patterning, the reversibility of the fluorescence of the TSFF under 254 nm UV light was also investigated. As shown by the results in Fig. 2d, the irradiation with 254 nm UV light deconstructs the dimer of the anthracene generated by the 365 nm UV irradiation and enriches the anthracene moieties in the TSFF. After successive irradiation with 254 nm UV light, the fluorescence intensity of the TSFF increases but cannot recover to its original level. As shown in Fig. 2d, a -10% loss of fluorescence appears after irradiation by 254 nm UV light for 4 h. This is because anthracene dimers are inherently hard to fully cleave, especially in a solid medium like the TSFF. Even so, the TSFF still holds a good reversible fluorescence-shifting behavior upon alternate irradiation by 365 and 254 nm UV light. In three successive cycles of alternating 365 and 254 nm UV irradiation, the TSFF maintains a good reversibility of fluorescence-shifting and the final fluorescence reaches 76% (Fig. 2e). The reversible fluorescence-shifting of the TSFF was then demonstrated visually based on the fluorescent images under a 254 nm UV light. As shown in Fig. 2f, ten samples taken from the same TSFF were divided into two groups; the five samples in the first row were exposed to 365 nm UV light for different times (0–2 h), and there is a distinct change of fluorescence from strong to weak emission; the other five samples were exposed to 365 nm UV light for 2 h firstly and then irradiated by 254 nm UV light for different times (0–4 h),

the fluorescence is enhanced as the bright fluorescent images returned.

From the perspective of 2D-encoding for information storage, the sharp contrast in the fluorescence of the TSFF induced by the alternate 365 nm and 254 nm UV light irradiation can benefit the creation of 2D patterns by photo-imaging; moreover, the reversible nature of the fluorescence-shifting can make the 2D information rewritable by repeatedly erasing and recreating the patterns.

Inspired by the above achievements, a premilary study was conducted on a TSFF for creating rewritable 2D information. Three representative mascots were chosen as photomasks: Bing Dwen Dwen (Olympic Winter Games Beijing 2022, pattern A), Rongbao (Chengdu 2021 FISU World University Games, pattern B), and Shuey Rhon Rhon (Beijing 2022 Paralympic Winter Games, pattern C). The 2D mascot patterns were expected to be printed on the TSFF using localized 365 nm UV irradiation and then erased using 254 nm UV irradiation (Fig. 2g). As demonstrated by Fig. 2h and Supplementary Fig. 5, patterns A, B, and C are sequentially created on the TSFF by the alternate UV irradiation and with the assistance of the certain mask. Naturally, prior to the creation of a new pattern by 365 nm UV irradiation, the TSFF was exposed to 254 nm UV light to erase the previous pattern. Notably, the sharp contrast in fluorescence in the areas with and without 365 nm UV light irradiation made the 2D patterns distinct; subsequently, the 254 nm UV light irradiation could erase the created patterns almost completely. Because the fluorescent images in Fig. 2f

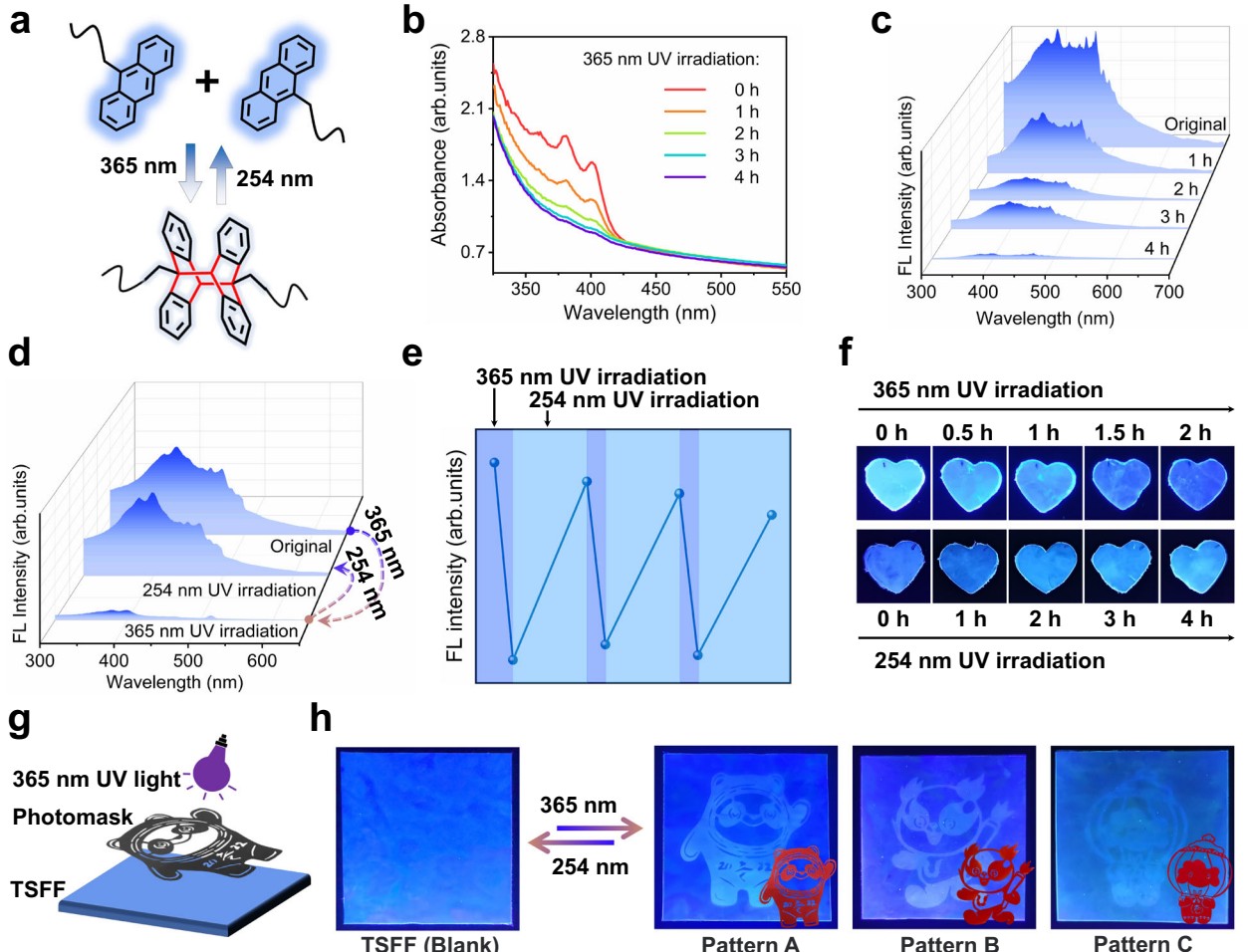

**Fig. 2 | Reversible fluorescence-shifting properties of TSFF. a** Reversible photo-cross-linking of anthracene groups. **b** UV-Vis spectra of TSFF after irradiating by 365 nm UV light for 0, 1, 2, 3 and 4 h. **c** Fluorescence spectra of TSFF after irradiating by 365 nm UV light for 0, 1, 2, 3 and 4 h. **d** Fluorescence spectra of TSFF after irradiating by 365 nm and 254 nm UV light for 0.5 and 2 h, respectively. **e** Change of fluorescence intensity upon alternate irradiation by 365 nm and 254 nm UV light for 0.5 and 2 h, respectively. **f** Visual demonstration of fluorescence change of two series of samples with an increasing irradiation time of 365 nm and 254 nm UV light. Ten heart-shaped samples were obtained from the same TSFF. Five of the ten

samples (first row) were simultaneously exposed to 365 nm UV light for different times (0–2 h), finally lined up and took the picture; the other five samples (second row) were irradiated with 365 nm UV light for 2 h and then exposed to 254 nm UV light for different times (0–4 h), finally the samples lined up and took the picture. **g** Schematic illustration of the creation of 2D fluorescent pattern achieved by irradiation with a 365 nm UV light with the assistance of photomasks. **h** Photos showing creation/erasure cycles of the 2D fluorescent patterns under 365 nm and 254 nm UV light. Pattern A: Bing Dwen Dwen, Pattern B: Rongbao, Pattern C: Shuey Rhon Rhon. Film size: 50 mm × 50 mm × 0.5 mm.

were taken by a camera, the distinct contrast of fluorescence was so strong that the areas not undergoing photo-cross-linking in the patterned images (A, B, C) were not the same as the blank TSFF, even though they looked the same with naked eyes; this is also the reason for the bright nature of the images in Fig. 2h. Moreover, the 2D fluorescent patterns created in this way had good stability; they remained distinguishable after storing the patterned TSFF in indoor ambient environment for 240 d or immersing in strong acid, strong base, or salt solution for 240 d, revealing the long-term effectiveness of information storage (Supplementary Figs. 6 and 7). In all, these results demonstrate that TSFF is capable of the potential for rewritable information for 2D encoding.

**Thermadapt shape-memory properties**

The thermadapt shape-memory properties refers to the thermal reconfiguration of the permanent shape of an SMP by activating the dynamic bonds in the polymer network, also called the thermally induced plasticity. In this work, the chemical structure of the TSFF meets the requirements for both the thermally induced plasticity and thermo-responsive SME. This is because cEVA is a typical crystalline

polymer with an abundance of dynamic ester bonds, laying the foundation for 3D encoding, the creation of a 3D pattern by the thermally induced plasticity, and the subsequent shielding by the macroscopically shape programming via SME.

First, the thermal properties related to the SME of the TSFF were investigated using differential scanning calorimetry (DSC). As shown in Fig. 3a, the prepared TSFF has melting and crystallization temperatures ($T_m$ and $T_c$, respectively) of approximately 70 and 45 °C, respectively. In addition, the crystallization-melting transition is distinct. This implies that the TSFF can be deformed at $T_d > 70$ °C and fixed at $T_f < 45$ °C. As shown in Fig. 3b, a strip TSFF sample (as the original shape, $S_o$) can be programmed into a temporary shape ($S_t$) by deforming it at 75 °C, fixing it at -25 °C, and finally recovering at 75 °C. These procedures are repeatable, leading to a cyclic SME involving different $S_t$.

Then, we aimed to demonstrate the thermally induced plasticity of the TSFF, which was expected to make the original shape of the TSFF reconfigurable. This advantage would make sense for the creation of 3D patterns standing for certain information. As reported, the dynamic nature of ester bonds can be reflected by their stress relaxation

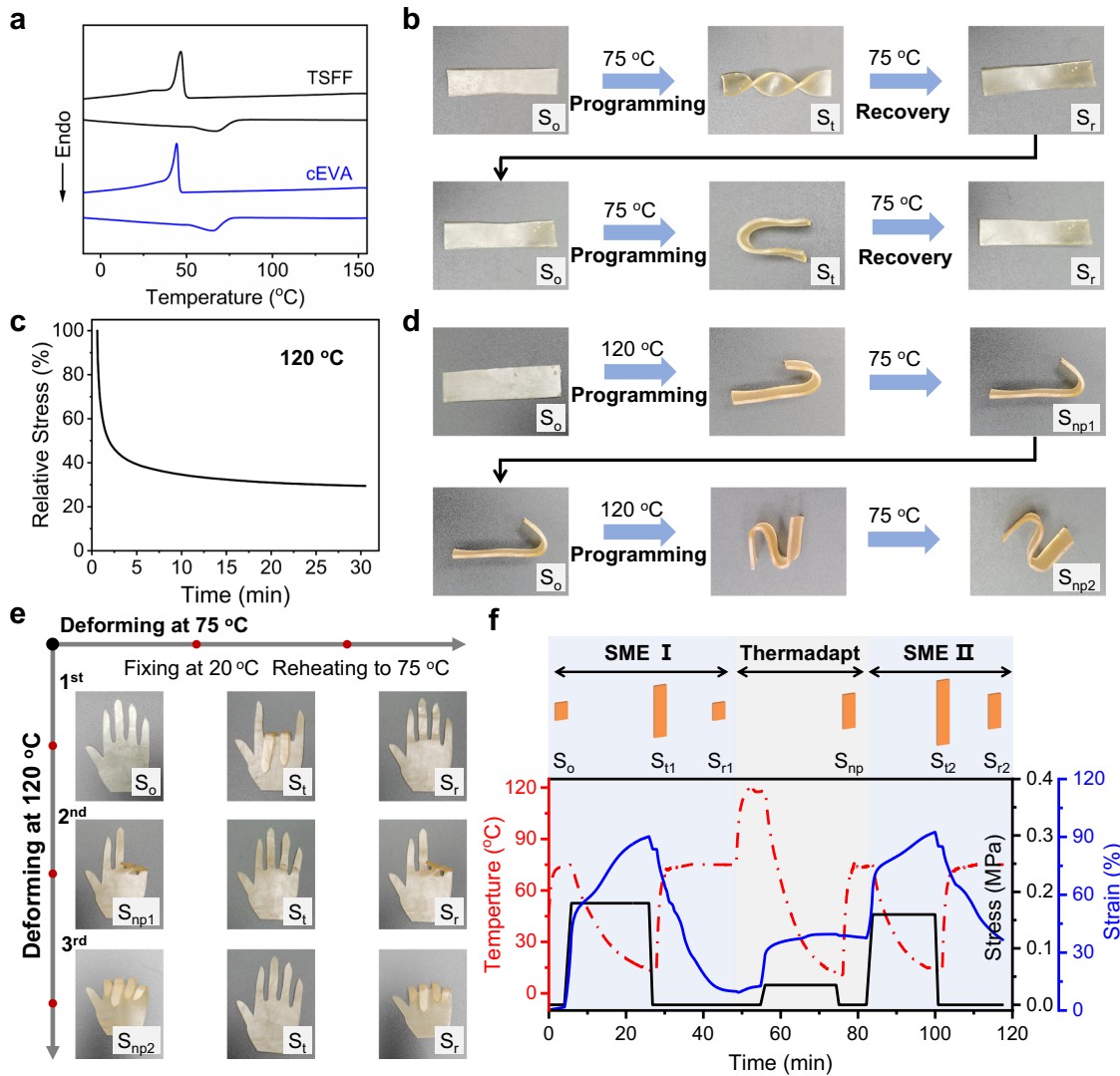

**Fig. 3 | Thermadapt shape-memory properties of TSFF. a** DSC results of TSFF and pristine cEVA. **b** Demonstration of the thermo-responsive SME of TSFF. **c** Stress relaxation of TSFF at 120 °C. **d** Demonstration of thermally induced plasticity of TSFF. **e** Demonstration of the permanent shape-reconfiguration and SME of TSFF. **f** Thermadapt shape-memory properties of TSFF as measured by DMA. $S_o$, $S_t$, and $S_r$ are defined as the original, temporary, and recovered shapes, respectively, and $S_{np}$ is defined as the new permanent shape produced by thermally induced plasticity.

behavior at a certain temperature. Herein, we examined the stress relaxation of the TSFF at a constant strain of 30% at 120 °C with the assistance of dynamic mechanic analysis (DMA). The results are presented in Fig. 3c. As expected, the TSFF holds a fast stress relaxation within 2 min before the relaxion trend slows; the final relaxion degree reaches 70%, revealing a high efficiency in the network reconfiguration induced by the dynamic bonds. To explore the effect of the small amount of the incorporated PCL-AN component on the stress relaxation of the TSFF, a pristine cEVA sample was also tested; a similar result is observed (Supplementary Fig. 8).

The dynamic feature of the TSFF can endow it with the ability to reconfigure its shape, that is, the thermally induced plasticity. As shown in Fig. 3d, the TSFF ($S_o$: rectangular strip) is deformed at 120 °C and a new shape can be produced after cooling to ambient temperature and releasing the external stress. The most important aspect is whether the newly produced shape can be recovered at 75 °C or not; it is only when it does not recover that suggests thermally induced plasticity. The results demonstrate that the produced shape (e.g., S-shape) cannot recover to $S_o$ when heated to 75 °C (Supplementary Movie 1), thus, it should be defined as a new permanent shape ($S_{np}$). The advantages of thermally induced plasticity can be used to create

3D patterns (information) on a flat TSFF with the assistance of a template.

We also studied the thermadapt shape-memory properties combining the shape-configuration process and SME. As discussed before, in this work, the shape-configuration process of the TSFF allows for the creation of 3D information (pattern), whereas the subsequent SME is responsible for the 3D information-shielding (encoding) by shape programming (for example, folding). Figure 3e displays images taken over an entire thermadapt shape-memory cycle. The TSFF ($S_o$: palm) was heated at 120 °C and then programmed into a new shape ($S_{np}$: crooked fingers). This shape could be maintained after heating to 75 °C, indicating a successful shape reconfiguration. Then, the TSFF in $S_{np}$ was programmed into an unfolded palm at 75 °C, here, the palm was in $S_t$ in SME as it recovered to the folded state after reheating to 75 °C (Supplementary Movie 2). The thermadapt shape-memory properties were totally reproducible as additional $S_{np}$ and $S_t$ could obtain at certain temperatures (120 and 75 °C, respectively).

Meanwhile, the thermadapt shape-memory properties of the TSFF was quantitatively investigated using a DMA (Fig. 3f). The procedures were similar to those mentioned above except that parameters such as

the temperature, stress, and time were automatically controlled by the DMA. As shown in Fig. 3f, the shape fixation ratio ($R_f$) and recovery ratio ($R_r$) of the TSFF are 92.5% and 90.7%, respectively, indicating an excellent thermo-responsive SME. In particular, the generated $S_{np}$ from the thermally induced plasticity is maintained almost completely in the following SME cycle. In addition, since cEVA is a semi-crystalline polymer with a two-way SME, it will exhibit significant crystallization-induced elongation when cooling under external stress[37,39], so the deforming strain of TSFF keeps increasing during cooling. In sum, the TSFF with thermadapt shape-memory properties is a promising and affordable approach to the creation of 3D information (pattern) and its shielding for 3D encoding.

## Reprocessing and self-welding properties

Dynamic bonds have become a great source for the development of multifunctional polymers besides thermadapt SMPs. These bonds demonstrate attractive features such as reprocessing and self-welding. Considering the abundance of dynamic ester bonds in the TSFF, it could feasibly hold these two properties. From the perspective of an information carrier, reprocessing and self-welding may provide material sustainability and information diversity. The reprocessability of the TSFF should be contributed to the dynamic exchange of ester bonds, as this induces the adaptivity of the polymer network. The experiment for reprocessing included cutting a TSFF into pieces and then remolding as shown in Fig. 4a. The temperature for remolding

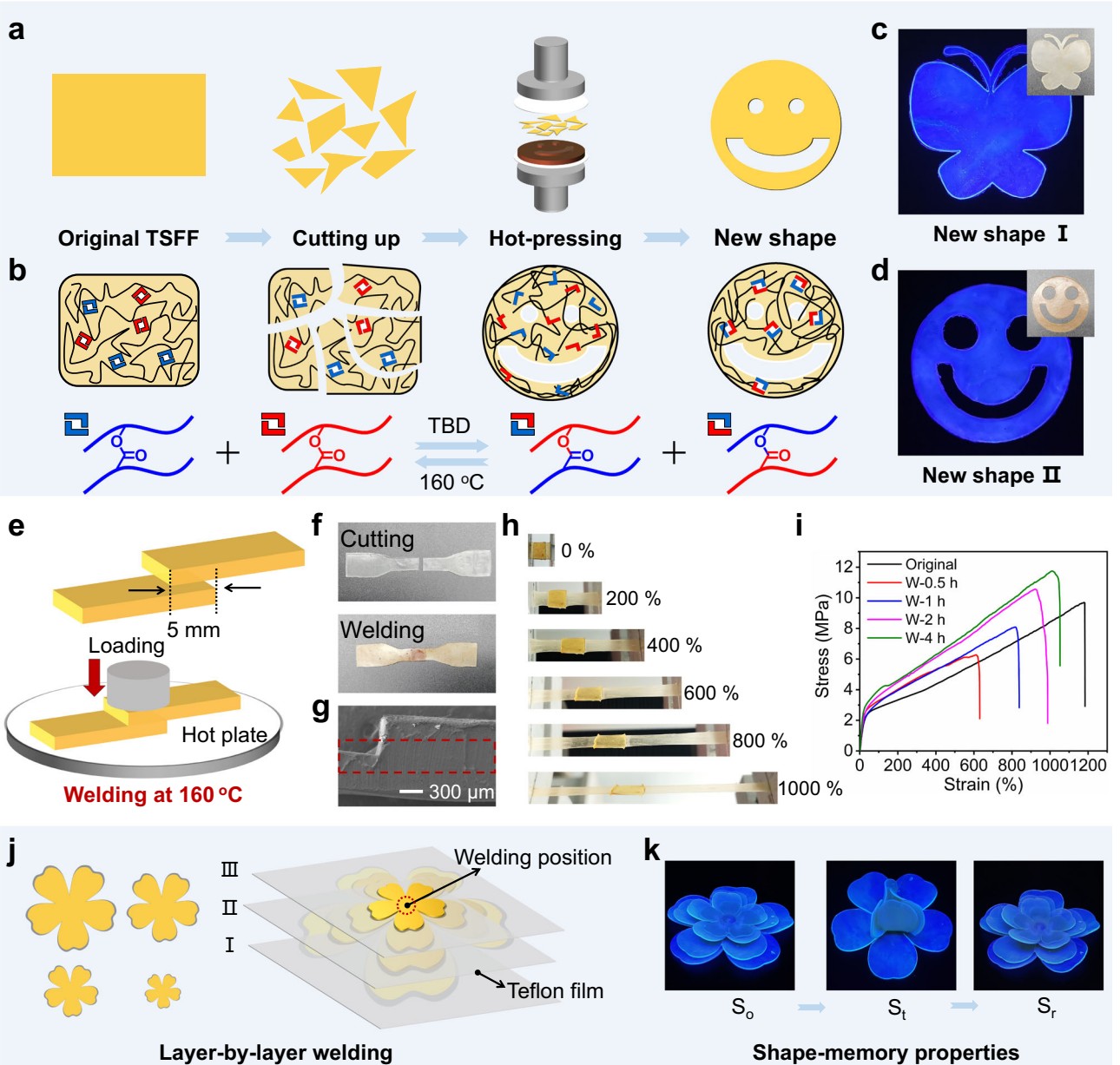

**Fig. 4 | Reprocessing and self-welding properties of TSFF. a** Illustration of reprocessing process. **b** Molecular mechanism of reprocessability. **c** Demonstration of the reprocessing of a sample into a butterfly shape. **d** Demonstration of the reprocessing of a sample into a smiley-face shape. **e** Schematic illustration of self-welding process. **f** Photographic image showing typical self-welding behavior: First, a dumbbell-shaped sample is cut into two halves by a blade and then welded at 160 °C for 4 h. **g** Scanning electron microscopy (SEM) image of a TSFF sample after welding. **h** Visual demonstration of stretching the welded TSFF sample to different strains (0%, 200%, 400%, 600%, 800%, and 1000%). **i** Stress-strain curves of TSFF samples at different times of welding (0.5, 1, 2, and 4 h). **j** Schematic illustration of assembling a 3D flower by welding 2D petals layer-by-layer. **k** Visual demonstration of the fluorescence and thermo-responsive SME of the welded 3D flowers.

was chosen as 160 °C to accelerate the dynamic exchange rate and the pressure for hot pressing was 5 MPa. The molecular mechanism is graphically illustrated in Fig. 4b, in which the dynamic nature of the ester bonds is the core. As a result, these pieces of the TSFF can be reprocessed into other shapes, such as butterflies and smiley faces (Fig. 4c, d). On the contrary, the controlled sample without adding TBD catalyst cannot be reprocessed (Supplementary Fig. 9), and this result confirmed the dynamic nature of TSFF in return. The mechanical properties of the samples before and after reprocessing were also studied using tensile testing. The results show that Young's modulus, tensile strength, and elongation at break of the reprocessed TSFF sample were almost identical to those of the original sample with only a very small variation (Supplementary Fig. 10), indicating the perfect formation of polymer networks. The perfect reprocessing of the TSFF was also demonstrated by the results from a comparative study on the chemical structure and fluorescence, both of which showed no significant change when compared with the original sample (Supplementary Figs. 11 and 12). The reprocessability of the TSFF provides significant benefits to its recycling and sustainability relative to the reported non-recyclable information materials (Supplementary Table 1).

Recently, self-welding has also been developed as an emerging property for extending the working lives of polymers and assembling complex geometries thereof. Owing to the abundance of dynamic ester bonds, theoretically, the TSFF can be made self-weldable by forming covalent bridges between the contact interfaces. Notably, the self-welding process is similar but not the same to reprocessing, because welding only occurs at the interfaces and not remolding, as graphically illustrated in Fig. 4e. To facilitate the investigation of the mechanical properties, a dumbbell-shaped specimen was prepared. After partly overlaying the two parts of the specimen and hot pressing, the two parts were fused together well (Fig. 4f and Supplementary Fig. 13). The resultant contact interfaces show a compatible topography (Fig. 4g) and the welded sample can lift a weight of 2 kg, i.e., 15000 times its own weight (Supplementary Fig. 14); both of these aspects demonstrate the successful welding of the TSFF.

To quantitatively investigate the self-welding efficiency, the mechanical properties of the welded TSFF were analyzed using tensile testing. Photos of the sample at different strains during the tensile testing are shown in Fig. 4h. They indicate that the welded TSFF does not break even after stretching to 1000%. As self-welding is related to the dynamic exchanges of the chemical bonds, the welding time influence the efficiency. Figure 4i summarizes the stress-strain curves of TSFF specimens with different welding times. The overall results indicate that a longer welding time results in higher efficiency of welding, as both the tensile stress and strain show positive correlations.

The self-welding process should be easy and not require molds, making it very useful for producing mold-free customized complex architectures and thereby enriching the geometry diversity of the information carrier. As shown in Fig. 4j, several 2D petals made from the TSFF are assembled into a 3D flower using layer-by-layer welding. The produced 3D geometry with independent layers is stable. The welded 3D flower still maintains fluorescence and SME, as demonstrated by Fig. 4k. This proof-of-concept for assembling complex geometries by the self-welding of the TSFF is similar to the traditional Chinese mortise and tenon technologies for interconnecting simple components for wooden buildings. Owing to the reversible fluorescence-shifting property of the TSFF, it can be used to create 2D patterns on each component of a structure, for example, the petals in this experiment. The natural light-invisible 2D pattern can be encrypted after folding by shape programming and decrypted after unfolding induced by shape recovery (Supplementary Fig. 15). Meanwhile, it was feasible to weld several TSFF pieces standing for practical 3D information on a TSFF substrate, the encryption/decryption procedures

were the same (Supplementary Fig. 16). These achievements proved the benefits of the self-welding property on information storage as the proof-of-concepts were similar to the origin of 2D/3D-encoding, which will be discussed more fully below.

## Rewritable 2D/3D encoding

The previous sections described the reversible fluorescence-shifting and thermadapt shape-memory properties of the TSFF. These achievements inspired us to investigate the feasibility of the TSFF as a rewritable 2D/3D-encoding information carrier. Prior to conducting a proof-of-concept of the rewritable 2D/3D-encoding, we first examined the 2D encoding of the TSFF. In fact, creating 2D images that are usually invisible under natural light but visible under UV light on a material surface is the most common way to realize anti-counterfeiting, for example, with paper currencies. Therefore, we assumed that the TSFF was a good candidate for anti-counterfeiting labels for information encryption because its 2D patterns are only visible under UV light.

As shown in Fig. 5a, two types of information (QR code and message) are printed on two pieces of TSFF by 365 nm UV light irradiation under photomasks, and the TSFF pieces are adhered to a gift box as labels. Figure 5b shows a real scene, the QR code is invisible under natural light, so it is impossible to scan the code to obtain the information; once the label is exposed to UV light (254 nm), the QR code becomes visible and a cellphone can scan it to immediately decode the information "NEVER GIVE UP." Similarly, a message consisting of a few sentences can also be decoded under UV light, as demonstrated in Fig. 5c. Moreover, as the TSFF is flexible, the derived labels can adapt to certain curved surfaces. For example, a TSFF label with the fluorescent pattern of the Sun and Immortal Birds (the logo of Chengdu City) can be attached to a glass bottle, with decoding procedures similar to those mentioned before (Fig. 5d). Finally, as a proof-of-concept for integrating multiple encrypted information on an existing medium, pieces of TSFF printed with different 2D information are adhered to a banknote model (Fig. 5e). There are four types of 2D information comprising the entire anti-counterfeiting code: letters, panda, numbers, and barcode. Only when all of them are decoded can the banknote be verified.

In sum, the 2D-encoding of TSFF by creating cryptographic information that can only be decrypted under UV light can enhance the security level of information storage. As the decoding of these fluorescent patterns is simple and can usually be read by naked eyes under a portable UV light, a risk of information leakage and counterfeit remains. In this context, the integration of 2D and 3D encoding in one system is of great significance. Moreover, providing rewritable 2D/3D-encoding makes these information carriers reusable.

Herein, our ultimate goal was to engineer the TSFF as an advanced information carrier with improved security as enabled by the integration of 2D and 3D encoding. The TSFF carrying 2D and 3D information can be folded by shape programming. Only upon the activation of the shape recovery can the 3D information be identified by the naked eyes; then, the 2D information be verified upon exposure to UV light. The two-stage verification process can greatly improve the security level of information because the designated inspectors not only know the existence of the 2D and 3D information, but also understand how to decode the true meanings that they stand for. Moreover, we have carefully considered the information input sequence that creates 3D pattern first and then 2D information, because anthracene dimers would be de-cross-linked at elevated temperatures[40], raising a risk of wiping the 2D fluorescent pattern if the sequence is reversed. Taking Fig. 6a as an example, the conceptual demonstration of 2D/3D encryption and decoding is as follows. First, a TSFF not bearing any information (Stage I) is printed with a 3D pattern by the thermally induced plasticity, i.e., the Olympic Rings (Stage II). As the 3D information is visible both under natural and UV light, it is not encrypted.

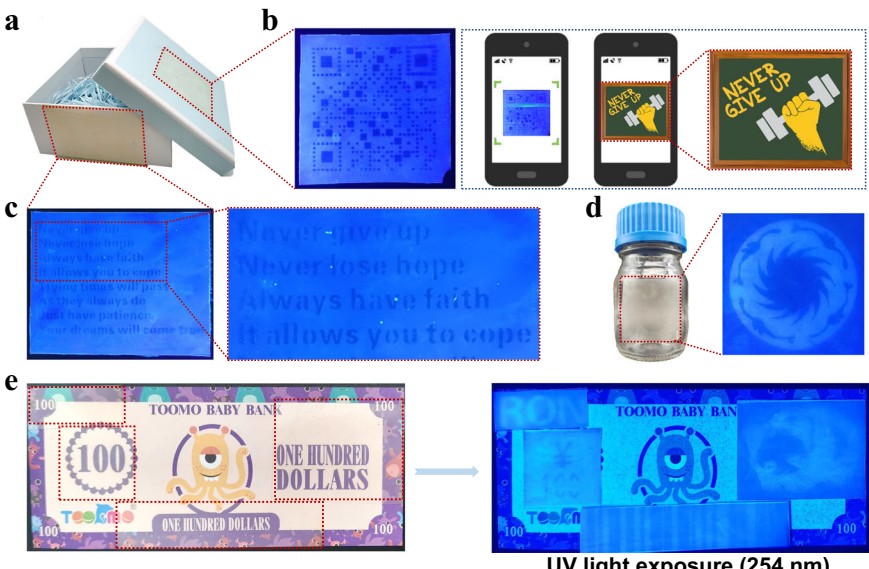

**Fig. 5 | 2D-encoding anti-counterfeiting strategy. a** Two pieces of TSFF carrying 2D information are adhered to a gift box as labels. **b** QR code visible under 254 nm UV light and decoded information. **c** A few sentences visible upon exposure to 254 nm UV light. **d** TSFF label attached to a glass bottle. The fluorescent pattern is the Sun and Immortal Birds (the logo of the city of Chengdu). **e** Integration of multiple encrypted information into banknote model on TSFF. Top left: letter RON; right: panda pattern; bottom left: ¥100; middle: barcode pattern.

Then, the emblem of the 2022 Beijing Olympic Winter Games is printed on the TSFF as 2D information by photo-patterning (Stage III); the 2D information is invisible under natural light but visible when exposure to UV light; thus, it is encrypted. Finally, to encrypt the former 3D information using the SME, the flat TSFF is programmed into a folded airplane-like shape (Stage IV); then, both the 2D and 3D information are sheltered inside and cannot be detected. After the multi-stage 2D/3D encoding, all of the 2D and 3D information can be identified by designated inspectors by heating the TSFF for shape recovery and exposing it to UV light in sequence (Stage V).

The light-induced reversible fluorescent-shifting and thermadapt shape-memory properties of the TSFF further enable the rewriting of the 2D and 3D information, making the TSFF a reusable carrier for on-demand information storage and encryption. As shown in Fig. 6b, the 2D and 3D information created in Fig. 6a can be erased completely by thermally induced plasticity (120 °C, 10 min) and de-cross-linking (254 nm UV irradiation, 2 h), this enables the rewriting of information; the true message "SWJTU" (the abbreviation for Southwest Jiaotong University) is displayed using Morse code (3D information) and characters (2D information), and the TSFF is programmed into a scroll to hide the information. Only when both the 2D and 3D information are detected can the true message "SWJTU" be verified. Moreover, Fig. 6c is an additional proof-of-concept of rewritable 2D/3D encoding, wherein the entire message is also composed of 2D and 3D information.

In sum, the TSFF holds the advantage of the rewritable 2D/3D encoding, which is benefited from the reversible fluorescent-shifting and thermadapt shape-memory properties; this feature can make the TSFF more versatile and multifunctional relative to other reported anti-counterfeiting materials (Supplementary Table 1).

## Discussion

In this work, we engineered TSFF with reversible fluorescence-shifting and thermadapt shape-memory properties as rewritable 2D/3D encoding information carriers with high security. The two functional components of the TSFF were the anthracene groups simultaneously bearing the reversible photo-cross-linking and fluorescence-shifting and the cEVA containing the abundant dynamic ester bonds. The

results showed that the fluorescence of the TSFF could be regulated by alternating UV light irradiation (365 and 254 nm), and that this process was repeatable. As such, with the assistance of certain photomasks, a diversity of 2D patterns can be printed on the TSFF, and these 2D patterns were fully erasable and rewritable on-demand. Meanwhile, the 2D patterns were invisible under natural light and only became visible under UV light (254 nm). This feature was applicable to simple 2D encoding, for example, in labels with QR codes and messages as demonstrated. The dynamic nature of the TSFF led to thermadapt shape-memory properties which not only enabled the reconfiguration of permanent shapes by activating the dynamic exchange of the ester bonds at high temperatures (120 and 160 °C, confirmed in this work), but also enabled conventional shape programming and recovery (at 75 °C). The former was demonstrated as useful for the creation and erasure of 3D patterns (rewritable), whereas the latter was beneficial to the sheltering of the produced 3D patterns. The combination of them led to the realization of 3D encoding. The integration of rewritable 2D/3D encoding can improve the security level of information because only the designated inspectors can understand the existence of 2D and 3D information and how to decode them. This was demonstrated by the proof-of-concept approach of heating the TSFF for shape recovery followed by UV exposure. The dynamic nature of TSFF also endowed it with reprocessing and self-welding properties, making it a multi-functional platform for information carriers. Overall, this TSFF capable of rewritable 2D/3D encoding will inspire the design of smart materials for high-security information carriers.

## Methods
### Materials

Poly (ethylene-co-vinyl acetate) (EVA, 28 wt% vinyl-acetate) was purchased from DuPont Packaging & Industrial Polymers (USA). Acryloyl chloride was purchased from Aladdin Reagent Co., Ltd. (Shanghai, China). Toluene, dichloromethane (CH$_2$Cl$_2$), ethanol, potassium carbonate (K$_2$CO$_3$), and calcium hydride (CaH$_2$) were purchased from Kelong Reagent Corp. (Chengdu, China). Stannous octoate (Sn(Oct)$_2$, 95.0%) and 9-Anthracenemethanol were purchased from Adamas Reagent Co., Ltd. (Shanghai, China). Dicumyl peroxide (DCP) and 1,5,7-triazabicyclo [4.4.0]-dec-5-ene (TBD) were purchased from Tianjin

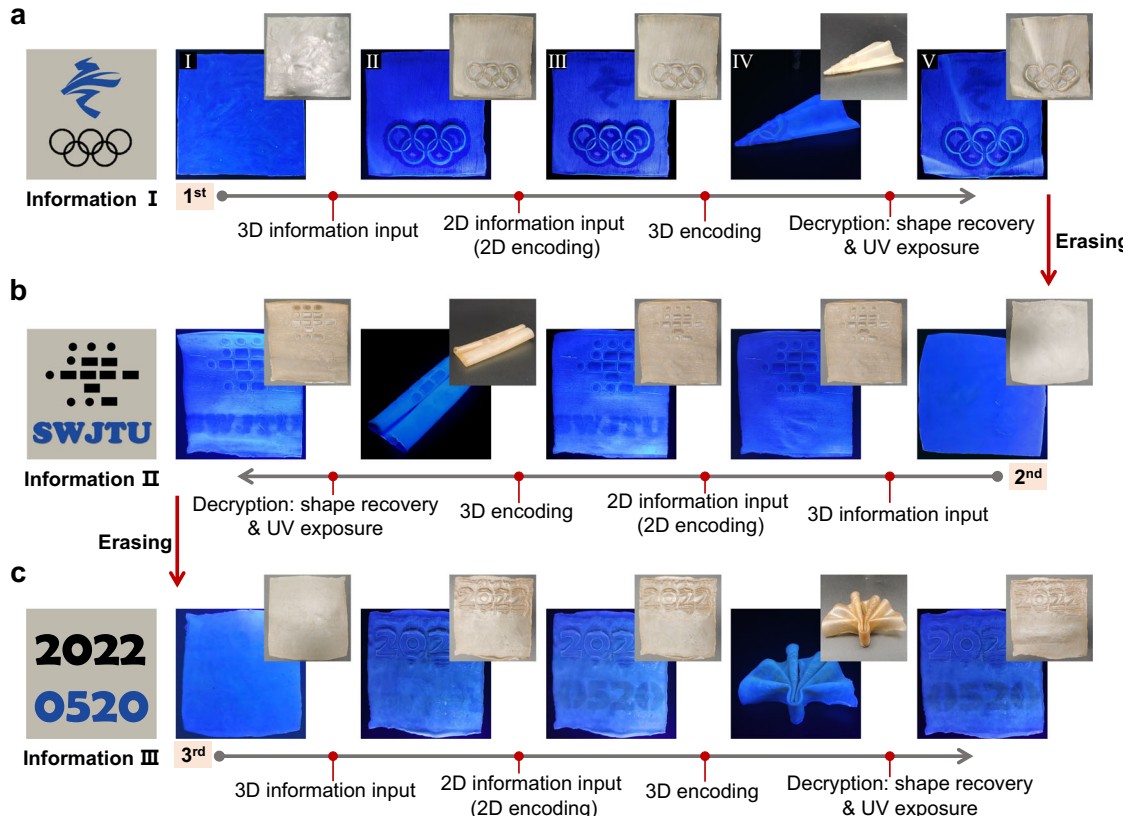

**Fig. 6 | Rewritable 2D/3D encoding for information encryption. a** The sequential input of 3D and 2D information (the Olympic Rings and the emblem of the 2022 Beijing Olympic Winter Games, respectively) by thermally induced plasticity and photo-patterning (Stage I–III), respectively. 2D-encoding is achieved after photo-patterning as the fluorescent pattern is invisible under natural light, while 3D-encoding is realized by shape programming of TSFF from flat to an airplane-like shape (Stage IV). The encrypted 2D and 3D information can be decoded after shape recovery of TSFF induced by heating and UV exposure (254 nm). **b** The second recreation of 3D and 2D information on TSFF after erasing the information in the first cycle by thermally induced plasticity (120 °C, 10 min) and de-cross-linking (254 nm UV irradiation, 2 h). The procedures are the same as those in the first cycle. In particular, the 3D information is the Morse code standing for SWJTU, the 2D information is message "SWJTU," and the shape for 3D encoding is scroll. **c** Additional demonstration of rewritable 2D/3D encoding. The procedures are the same as mentioned above. Here, the 3D information, 2D information, and the shape for 3D encoding are "2022," "0520," and a folding fan, respectively.

Heowns Biochemical Technology Co., Ltd. (Tianjin, China). ε-Caprolactone was purchased from Alfa Aesar Chemical Co., Ltd. (Shanghai, China).

### Synthesis of acryloylated anthracene-containing poly-caprolactone (AC-PCL-AN)

First, the PCL-AN was prepared by the ring-opening polymerization of ε-Caprolactone by 9-Anthracenemethanol. The ε-Caprolactone was purified by distillation under reduced pressure after adding $CaH_2$ and stirring at room temperature for 12 h. 9-Anthracenemethanol (1.89 g, 9.08 mmol) was added to a three-necked flask and dried under a vacuum at 90 °C for 2 h. Then, the purified ε-Caprolactone (17.0 mL, 159 mmol) and 3–4 drops of $Sn(Oct)_2$ were added in turn, and the reaction was carried out under nitrogen atmosphere at 140 °C for 12 h. After cooling to ambient temperature, solid product (PCL-AN) was obtained.

Second, the obtained PCL-AN product was modified by acryloyl chloride. The PCL-AN product was dissolved in anhydrous $CH_2Cl_2$ (50 mL), and $K_2CO_3$ (6.35 g, 45.9 mmol) was added as the acid-binding agent. The reaction solution was stirred under ice bath conditions for 0.5 h. Then, acryloyl chloride (3.68 mL, 45.3 mmol) was dissolved in anhydrous $CH_2Cl_2$ (10 mL) and added dropwise to the reaction mixture. The reaction was stirred for 48 h while maintaining the ice bath conditions. After the reaction was completed, the reaction mixture was concentrated to 20 mL by rotary evaporation and then precipitated by cold ethanol. Finally, the resulting precipitate was filtered under reduced pressure. After dried to a constant weight under vacuum at 25 °C, 14.6 g of yellow solid product (AC-PCL-AN) was obtained.

### Preparation of TSFF

The EVA pellets (8.93 g) were dissolved in toluene (100 mL) at 65 °C, and then AC-PCL-AN (0.47 g), TBD (0.3 g) and DCP (0.3 g) were added and stirred for 2 h. After fully dissolving, the mixture was transferred to a glass dish to remove the solvent at room temperature for 24 h and further dried under vacuum. Finally, the obtained product was placed in a rectangular mold (size: 60 mm × 40 mm × 0.5 mm), and films (TSFF) were obtained after molding by hot-pressing at 100 °C and then cross-linking at 170 °C for 20 min.

### Investigation of reversible fluorescence-shifting properties

The experiments were performed by irradiating the TSFF sample with UV light (365 and 254 nm) for different amounts of time. The changes in the fluorescence intensity from 300 to 700 nm were recorded by a spectrofluorometer (Horiba, USA). For the study of the reversible fluorescence-shifting, a TSFF sample was alternately irradiated by 365 nm UV light for 0.5 h and 254 nm UV light for 2 h. The entire process was repeated three times and the fluorescence intensity of the TSFF was recorded at each stage.

### Investigation of thermal properties

The melting temperature ($T_m$) and crystallization temperature ($T_c$) of the TSFF sample were characterized using a differential scanning

calorimetry (DSC-2500, TA, USA). The sample was rapidly heated to 160 °C and kept for 3 min to eliminate thermal history. Subsequently, the sample was cooled to −20 °C and reheated to 160 °C in successive cycles at a rate of 10 °C/min.

## Investigation of thermadapt shape-memory properties

The thermadapt shape-memory properties of TSFF were measured using a dynamic mechanical analyzer (DMA-Q800, TA, USA). The entire procedure was divided into three cycles. For the first cycle, the sample was heated at 75 °C for 10 min and maintained an initial strain of $\varepsilon_0$. Next, the sample was stretched by applying stress and then cooled to 10 °C; at this point, it had a deformed strain of $\varepsilon_d$. After releasing the stress, a temporary shape with a strain of $\varepsilon_f$ was achieved. Finally, the sample was reheated to 75 °C and recovered to its original shape with a residual strain of $\varepsilon_r$. In the second cycle, the sample was heated to 120 °C and maintained for 5 min. Next, the permanent shape reconfiguration was achieved by applying stress to stretch the sample, cooling to 10 °C, and releasing the stress. Finally, the steps in the first cycle were repeated for verifying the SME after permanent shape reconfiguration (the 3rd cycle). Based on result in the first cycle, the shape fixity ratio ($R_f$) and recovery ratio ($R_r$) were calculated according to Eqs. (1) and (2), respectively.

$$R_f = \frac{\varepsilon_f - \varepsilon_0}{\varepsilon_d - \varepsilon_0} \times 100\% \qquad (1)$$

$$R_r = \frac{\varepsilon_f - \varepsilon_r}{\varepsilon_f - \varepsilon_0} \times 100\% \qquad (2)$$

## Investigation of reprocessing property

A rectangular TSFF sample was cut into pieces, placed on a mold (e.g., butterfly-shaped), and hot-pressed at 160 °C with a pressure of 5 MPa. These processing procedures were repeatable, other shapes (e.g., smiley-face-shaped) could be obtained.

## Investigation of self-welding property

First, a standard dumbbell-shaped TSFF sample (50 mm × 8.5 mm × 0.5 mm, the length and width of the narrow section were 16 mm and 4 mm, respectively) was cut into two pieces from the middle by a scalpel, and the top and bottom surfaces were brought into contact. Then, the samples were welded by hot-pressing at 160 °C for 0.5, 1, 2, and 4 h, the pressure was 5 MPa. The morphologies of the weld interfaces were qualitatively assessed using scanning electron microscopy (S3500, Hitachi Instruments, Japan) at an accelerating voltage of 5–10 kV. The samples were fractured by liquid nitrogen and sputtered with gold before testing.

## Mechanical properties

The mechanical properties of the original, reprocessed, and self-welded TSFF samples were investigated by tensile tests using a universal testing machine (HZ1004, Lixian Instrument Technology, China). Standard dumbbell-shaped samples (50 mm × 8.5 mm × 0.5 mm, the length and width of the narrow section were 16 mm and 4 mm, respectively) were prepared at first. The tensile tests were carried out at room temperature and at a strain rate of 60 mm/min, strain-stress curves were recorded. The data on Young's modulus, tensile strength, and elongation at break were collected for comparative study.

## Other characterizations

$^1$H NMR spectra of PCL-AN and AC-PCL-AN were obtained by AVANCE NEO 400 MHz spectrometer (Bruker, Germany) at room temperature using $CDCl_3$ as the solvent and tetramethylsilane as the internal reference. The FT-IR spectra of all samples were recorded by Nicolet 5700 instrument (Nicolet, USA), the scans were performed 32 times in the range of 4000–500 $cm^{-1}$, and the resolution was 4 $cm^{-1}$. The UV-Vis absorption spectra of TSFF were recorded by a UV-2550 spectrometer (Shimadzu, Japan) in the wavelength range of 200–600 nm, the sample size was 40 mm × 10 mm × 0.5 mm.

## Data availability

All data supporting the findings of this study are available within the article, as well as the Supplementary Information file, or available from the corresponding authors upon reasonable request. Source data are provided with this paper.

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

## Acknowledgements

This work was supported by the National Natural Science Foundation of China (52203188 to H.X.), the Sichuan Science and Technology Program (2020YFH0058, 2022YFSY0017 to H.X.), the New Interdisciplinary Cultivation Funds of Southwest Jiaotong University (2682023JX005 to H.X.), and the Fundamental Research Funds for the Central Universities (2682022ZTPY039 to H.X.).

## Author contributions

H.X. and S.Z. proposed and conceived the project. J.H. performed all the experiments and analyzed the data. Y.J. and Q.C. assisted in polymer synthesis. J.H. and H.X. wrote and revised the paper. H.X. and S.Z. supervised the research. All authors read and approved the final version of the manuscript.

## Competing interests

The authors declare no competing interests.
