## [Peer review file · Nature Communications]

REVIEWER COMMENTS

Reviewer #1 (Remarks to the Author):

This work by Huang et al. describes a kind of thermadappt shape-memory fluorescent film (TSFF) for the development of novel information carriers capable of rewritable 2D- and 3D-encoding. This work presents a new strategy for ensuring the high-security of information carriers by integrating 2D encoding based on light-triggered fluorescence-shifting and 3D encoding based on thermadappt shape-memory effect in one system. Meanwhile, the derived information carriers can be rewritable and sustainable due to the light- and thermally-induced dynamic bonds in TSFF. Overall, the results presented in the manuscript can support the idea of rewritable 2D- and 3D-encoding, and it can inspire the development of high-security information carriers from dynamic materials. Thus, I recommend this manuscript for publication in Nature Communications, and the authors should address the following comments.

1. The authors claimed that the mimic octopus capable of changing colors and shapes was the inspiration for the current work, but I did not see any details about the mimic octopus. To facilitate the broad readers of the journal, the authors should add more content about this aspect. Here, a picture that illustrates the mimic octopus and its color/shape-shifting is a recommendation.
2. The rewritable 2D encoding of TSFF is enabled by the reversible photo-cross-linking of anthracene moieties, which is accompanied by a repeatable fluorescence-shifting behavior. The results in Figures 2 and 5 support this point well but remain confusions. First, in Figure 2, why the irradiation time for 254 nm UV light was much longer than that for 365 nm UV light? Second, can a cellphone scan the QR code printed on TSFF? Figure 5b looks like a prototype, not a real scenario.
3. The description of thermadappt shape-memory effect may frustrate readers. What does the recoverable shape programming refer to? In addition, although the pictures in Figure 3 reveal the TSFF holds thermadappt shape-memory effect qualitatively, I still noticed that the stretched sample showed an obvious elongation during cooling, even the temperature was well below its T_c , and no plateau was achieved in the DMA test (Figure 3f). The authors should explain this phenomenon. Besides, the schematic of the thermadappt shape-memory effect (Figure 3f) is not in accordance with the strain curve recorded by DMA, for example, the S_{np} cannot be obtained before the removal of the external stress.
4. The reprocessing and self-welding properties are interesting, and they provide the TSFF with sustainability and shape-diversity add-values. Here, the first question for this part is how to weld multi-layers at the same position (Figure 4j). The welding position at the central seems to be too small to precisely heat. Second, how many times the experiment on the reprocessing property was carried out (Figure 10 in SI)? The authors should repeat the reprocessing experiment several times and then compare the mechanical properties of the reprocessed sample with that of the original one. Thus, additional experiments should be performed to evaluate the reprocessing property of TSFF.
5. Is there a certain sequence for the input of 3D and 2D information? The authors created 3D patterns (information) by thermal plasticity first, followed by printing 2D images by spatiotemporal light irradiation (Figures 1 and 6), I wonder whether these two processes can be reversed.
6. A minor comment: merging some of the supplementary movies into one is better.

Reviewer #2 (Remarks to the Author):

The manuscript entitled "Bioinspired Thermadappt Shape-Memory Polymer with Light-Induced Reversible Fluorescence for Rewritable 2D/3D-Encoding Information Carriers" described a thermally programmable shape-memory polymer materials with UV responsive fluorescence for data encryption and decryption. The shape memory effect was achieved through the thermal dynamic covalent bonds, while reversible fluorescent was achieved via anthracene motifs that undergoes photo-dimerization under UV irradiation. This polymer material shows double information encryption. The results are

interesting. Revisions are required before the manuscript can be considered for publication.

1) Generally, the innovative contributions of this paper required to be redefined and clarified throughout the paper. The material systems used in this manuscript, including shape memory effect of EVA and light-tunable fluorescence of anthracene motifs, have been studied in literatures. Introducing the concept of octopus biomimicry alone may risk exaggerating the function of the materials.

2) The reason for introducing the transesterification in EVA is not clear. If the objective is to introduce a permanent deformation, is it possible to design more complex multi-step deformations combining with crystallization/melting? Similarly, if the goal is to achieve reprocessability and self-welding, the author should provide more data on vitrimer properties rather than solely reporting the experimental results.

3) Anthracene was employed as a tunable fluorescent moiety, which was often used for modulating mechanical properties and achieving shape memory effects. The authors need to provide further discussion regarding the influence of the photo-dimerization of anthracene on the mechanical properties and shape memory effect. Also, does the formation of covalent bonds through photo-dimerization decrease the reprocessability of the material?

4) The disassociation of anthracene dimers can also be thermally induced. When programming the shape by heating, did the fluorescence pattern get affected?

5) In Figure 4i, why the welded specimen shows higher stiffness than the pristine one?

Reviewer #3 (Remarks to the Author):

This work reports a kind of functional polymers with thermadappt shape-memory effect and reversible fluorescence-shifting properties for high-security information carriers. The idea that integrates 2D and 3D encoding in a single carrier is interesting, and this is significant for improving information security. The authors also confirm that the reversible photo-cross-linking and thermally-induced plasticity can make both 2D and 3D information rewritable, this endows the derived information carriers with reusability. The experiments, results, and discussions presented in the manuscript are clear. In all, this work will inspire the development of multifunctional and high-security information carriers, I recommend it for publication in Nature Communications. However, the authors should clarify the following issues clearly.

1. In the Introduction, the authors claimed that a hydrogel-based information carrier is susceptible to dehydration, what the situation would be if the hydrogel has been designed to be anti-dehydration? Meanwhile, what about the long-term stability of the TSFF in the current work?

2. The authors stated that the shape memory effect in the previous works was merely used for information encasement and not for practical information (Line 60), so this work proposes TSFF with a thermadappt shape memory effect. The authors did not clearly state why the thermadappt shape memory effect can address these concerns. In addition, the authors should keep the consistency of expression with respect to “thermadappt” in each section.

3. The TSFF in this work holds the property of self-welding, as demonstrated in Figure 4, but the authors rarely discuss its connection with information storage, the core of this work. What benefits can the self-welding property bring to information anti-counterfeiting?

4. The pictures in Figure 4e showing the 2D information encoding of TSFF are not well matched, the areas (highlighted by red frames) camouflaged by several TSFF specimens are not consistent with those after exposure to UV light. For instance, the authors can compare the word “BANK” before and after UV exposure. The authors should carefully mark them to avoid any confusion.

5. As documented, photo-patterning is also a conventional approach to generate 3D geometries from a flat 2D film upon localized light irradiation, is there any possibility that the TSFF became 3D when precisely creating 2D surface patterns?

6. The synthesis route of AC-PCL-AN presented in Supplementary Figure 1 is not consistent with the details described in the Experimental section, and the chemical structure of AC-PCL-AN in Figure 2 should be checked accordingly.

Response to the reviewers' comments

For Reviewer #1:

This work by Huang et al. describes a kind of thermadappt shape-memory fluorescent film (TSFF) for the development of novel information carriers capable of rewritable 2D- and 3D-encoding. This work presents a new strategy for ensuring the high-security of information carriers by integrating 2D encoding based on light-triggered fluorescence-shifting and 3D encoding based on thermadappt shape-memory effect in one system. Meanwhile, the derived information carriers can be rewritable and sustainable due to the light- and thermally-induced dynamic bonds in TSFF. Overall, the results presented in the manuscript can support the idea of rewritable 2D- and 3D-encoding, and it can inspire the development of high-security information carriers from dynamic materials. Thus, I recommend this manuscript for publication in Nature Communications, and the authors should address the following comments.

Response: Thanks very much for your recommendation. We have carefully discussed your constructive comments and then made changes in the manuscript accordingly. The following is our point-by-point response to the comments.

1. The authors claimed that the mimic octopus capable of changing colors and shapes was the inspiration for the current work, but I did not see any details about the mimic octopus. To facilitate the broad readers of the journal, the authors should add more content about this aspect. Here, a picture that illustrates the mimic octopus and its color/shape-shifting is a recommendation.

Response: Thanks for your kind comment. Mimic octopus was the inspiration for us to develop information carriers with rewritable 2D/3D-encoding capacity. Previous works were usually focused on the color-changing behavior of octopus, which has inspired the development of anti-counterfeiting platforms based on fluorescence-shifting behavior. This is the conventional 2D-encoding strategy.^{1, 2} In this study, we aimed to develop information carriers with high-security and stumbled upon a special kind of octopus,

mimic octopus, a smart aquatic animal capable of changing both color (2D) and shape (3D) to adapt to the surrounding environment. It was inspired by the mimic octopus that we developed and engineered the thermadapt shape-memory fluorescent films (TSFF) as information carriers capable of 2D/3D-encoding.

We realized that more content on the mimic octopus should be added to facilitate broad readers to understand the shape/color-shifting behavior. Therefore, by following the reviewer's suggestion, we presented a schematic diagram showing a mimic octopus disguised as a coral to escape from predators by changing both shape and color, please see Fig. R1-1 (Supplementary Fig. 1 in the Supplementary Information). We think readers can get the idea of this work from this picture and the relevant description in the Introduction, *“Herein, inspired by the mimic octopus, one of nature's camouflage masters capable of changing its color and shape to adapt to the surrounding environment or escape from predators (Supplementary Fig.1), we designed and fabricated a type of thermadapt shape-memory fluorescent film (TSFF) using commercial poly (ethylene-co-vinyl acetate) (EVA) and anthracene for a rewritable 2D/3D-encoding information carrier (Fig. 1).”* (Line 85-89, Page 4, in the revised manuscript)

Fig. R1-1 (Supplementary Fig. 1 in the Supplementary Information) Illustration of the biomimicry of the mimic octopus, which camouflages as a coral by shape/color-shifting.

References:

1. Shi, H., et al. Cephalopod-inspired design of photomechanically modulated display systems

for on-demand fluorescent patterning. *Adv. Mater.* **34**, 2107452 (2022).

2. Lu, W., Si, M., Le, X. & Chen, T. Mimicking color-changing organisms to enable the multicolors and multifunctions of smart fluorescent polymeric hydrogels. *Acc. Chem. Res.* **55**, 2291-2303 (2022).

2. The rewritable 2D encoding of TSFF is enabled by the reversible photo-cross-linking of anthracene moieties, which is accompanied by a repeatable fluorescence-shifting behavior. The results in Figures 2 and 5 support this point well but remain confusions. First, in Figure 2, why the irradiation time for 254 nm UV light was much longer than that for 365 nm UV light? Second, can a cellphone scan the QR code printed on TSFF? Figure 5b looks like a prototype, not a real scenario.

Response: Thanks for the good comment. In this work, the fluorescence-shifting behavior of TSFF was mediated by UV irradiation with different wavelengths (365 nm and 254 nm). As shown in Fig. 2a, the anthracene group can bear a reversible photo-cross-linking, when irradiating by 365 nm UV light, they form dimer, and further irradiating by 254 nm UV light leads to the disassociation of the formed dimers. We have studied the change of fluorescence intensity of TSFF upon irradiating with 365 nm and 254 nm UV light for different times, as shown in Fig. 2c, f. It can be found that the decrease in fluorescence intensity after 365 nm UV irradiation is larger than the increased amount after irradiating with 254 nm UV light for an equal time. Fig. 2e shows that TSFF has a large loss of fluorescence intensity in cyclic alternating 365/254 nm UV irradiation. This is largely attributed to the fact that anthracene dimers are intrinsically difficult to cleave completely, especially in a solid medium.³ Therefore, to achieve high reversibility, the irradiation times for 365 nm and 254 nm UV light were determined to be 0.5 h and 2 h, respectively. As demonstrated by the results in Fig. 2f and h, both the fluorescence-shifting and reversible photo-patterning behavior of TSFF are satisfied when irradiating with 365 nm UV and 254 nm UV light.

As for the explanation in Fig. 5b, the QR code was scannable by cellphone. The QR code was valid and the recognized content was authentic. In the manuscript, to ensure the aesthetics of the figures, we drew a schematic diagram to better illustrate the

recognition process of the QR code. Here, we have photographed the actual process for reviewers' reference (Fig. R1-2).

Fig. R1-2 A real scenario in which a cellphone recognizes a QR code on TSFF.

Reference:

3. Van Damme, J. & Du Prez, F. Anthracene-containing polymers toward high-end applications. *Prog. Polym. Sci.* **82**, 92-119 (2018).

3. The description of thermadapt shape-memory effect may frustrate readers. What does the recoverable shape programming refer to? In addition, although the pictures in Figure 3 reveal the TSFF holds thermadapt shape-memory effect qualitatively, I still noticed that the stretched sample showed an obvious elongation during cooling, even the temperature was well below its T_c , and no plateau was achieved in the DMA test (Figure 3f). The authors should explain this phenomenon. Besides, the schematic of the thermadapt shape-memory effect (Figure 3f) is not in accordance with the strain curve recorded by DMA, for example, the S_{np} cannot be obtained before the removal of the external stress.

Response: We sincerely thank the reviewer for these constructive comments.

First, recoverable shape programming refers to the shape-memory effect process, because the shape programming by thermally-induced plasticity is not recoverable. In this work, we created 3D patterns (information) by thermally-induced plasticity; this process may also be regarded as a shape programming process, but it should be the reconfiguration of permanent shape. The created 3D patterns can only be erased when

heating to the temperature that activates the exchange of the dynamic bonds. Then, we programmed the flat patterned TSFF into complicated shapes to hide both the 2D and 3D information, this is the shape programming process; as the programmed temporary shape can recover to the original shape upon heating and thus enable the verification of information, the former programming process is considered as recoverable. These two procedures constitute a complete shape-memory effect.

Second, the reason why the stretched TSFF sample showed an obvious elongation during cooling should be attributed to the fact that cross-linked EVA (cEVA) is a semi-crystalline polymer network with a two-way shape-memory effect (2W-SME). The 2W-SME of cEVA relies on the effects of melting-induced contracting (MIC) and cooling-induced elongation (CIE), usually under constant external stress.⁴⁻⁶ Therefore, in the DMA tests, the stretched TSFF sample showed a significant elongation during cooling. Meanwhile, the elongation of TSFF will be relatively slow at last, as demonstrated by Fig. R1-3, the strain rate is almost constant around 0 before unloading. As it took a long time to achieve a plateau, we went to the next procedures without longer waiting.

Fig. R1-3 Rate of strain change of samples during DMA testing.

Since this work only involves a one-way shape-memory effect, we would like not to discuss the 2W-SME in the manuscript. Anyhow, to clarify the strain elongation of TSFF during cooling, the following text has been added to the revised manuscript.

“In addition, since *cEVA* is a semi-crystalline polymer with a two-way SME, it will exhibit significant crystallization-induced elongation when cooling under external stress,^{37, 39} so the deforming strain of TSFF keeps increasing during cooling.” (Line 234-236, Page 11, in the revised manuscript)

Finally, we realized that the schematic of the thermadapt shape-memory effect is not in accordance with the strain curve recorded by DMA (Fig. 3f). We have checked the curves of strain and temperature, and the schematic has been reassembled carefully, as shown in Fig. R1-4 (Fig. 3f in the revised manuscript).

Fig. R1-4. (Fig. 3f in the revised manuscript) f Thermadapt SME of TSFF as measured by DMA. S_o , S_t , and S_r are defined as the original, temporary, and recovery shapes, respectively, and S_{np} is defined as the new permanent shape produced by thermally induced plasticity.

References:

- Li, J., Rodgers, W.R. & Xie, T. Semi-crystalline two-way shape memory elastomer. *Polymer* **52**, 5320-5325 (2011).
- Xie, H., Li, L., Cheng, C.-Y., Yang, K.-K. & Wang, Y.-Z. Poly (ethylene-co-vinyl acetate)/graphene shape-memory actuator with a cyclic thermal/light dual-sensitive capacity. *Compos. Sci. Technol.* **173**, 41-46 (2019).
- Gao, Y., Liu, W. & Zhu S. Reversible shape memory polymer from semicrystalline poly (ethylene-co-vinyl acetate) with dynamic covalent polymer networks. *Macromolecules* **51**, 8956-8963 (2018).

4. The reprocessing and self-welding properties are interesting, and they provide the TSFF with sustainability and shape-diversity add-values. Here, the first question for this part is how to weld multi-layers at the same position (Figure 4j). The welding position at the central seems to be too small to precisely heat. Second, how many times the experiment on the reprocessing property was carried out (Figure 10 in SI)? The authors should repeat the reprocessing experiment several times and then compare the mechanical properties of the reprocessed sample with that of the original one. Thus, additional experiments should be performed to evaluate the reprocessing property of TSFF.

Response: Thanks very much for the constructive comment. The following is our point-by-point response to the comment.

First, the welding of multi-layers of TSFF was assisted by Teflon films that punched a hole in the center. In fact, we made a great effort to present the welding process vividly, and after several attempts, we chose this method. As shown in Fig. R1-5 (Fig. 4j in the revised manuscript), four pieces of TSFF samples are separated by three Teflon films with a small hole in the center, so only the center point of the TSFF pieces can contact each other. Upon heating to activate the exchange of dynamic bonds, the contacted areas are welded by compression.

Fig. R1-5 (Fig. 4j in the revised manuscript) Schematic illustration of assembling a 3D flower by welding 2D petals layer-by-layer.

Second, according to the reviewer's comment, we carried out additional experiments on the reprocessing properties of TSFF. The results indicate that Young's modulus, tensile strength, and elongation at break have no significant change after three cycles (Fig. R1-6). We added the following text and figure to the revised manuscript.

“The results show that Young's modulus, tensile strength, and elongation at break of the reprocessed TSFF sample were almost identical to those of the original sample with only a very small variation (Supplementary Fig. 10), indicating the perfect formation of polymer networks.” (Line 260-263, Page 13, in the revised manuscript).

Fig. R1-6. (Supplementary Fig. 10 in the Supplementary Information) Comparison of the mechanical properties between the original and reprocessed TSFF samples.

5. Is there a certain sequence for the input of 3D and 2D information? The authors created 3D patterns (information) by thermal plasticity first, followed by printing 2D images by spatiotemporal light irradiation (Figures 1 and 6), I wonder whether these two processes can be reversed.

Response: Thanks for the good comment. In this work, the sequence of the input of 3D information and 2D information was carefully determined. As shown in Fig. 6, the 3D pattern (information) was first created by thermally-induced plasticity, and then the 2D fluorescence pattern was printed by photo-irradiating of 365 nm UV light and with the assistance of certain masks. The 2D patterns can be erased by irradiating with 254 nm UV light due to the de-cross-linking of anthracene dimers. As documented, the de-cross-linking of anthracene dimers can also be triggered by heating, usually at elevated temperatures.⁷⁻¹⁰ Actually, the temperature of thermally-induced plasticity to create 3D information is 120 °C, which may trigger the de-cross-linking of the anthracene dimers, that is, the 2D information may disappear. Therefore, the creation of 3D information is ahead of 2D information.

Moreover, by following the reviewer's comment, we did an experiment to reverse the sequence of input of 3D/2D information, that is, creating 2D fluorescence pattern

first, followed by the creation of 3D information at 120 °C. As shown in Fig. R1-7, the 2D fluorescence pattern (“0520”) became blurred after creating 3D pattern (“2022”) at 120 °C. Therefore, it is better to not reverse the input sequence of 3D and 2D information.

To make this clear, we have added the following text to the revised manuscript.

“Moreover, we have carefully considered the information input sequence that creates 3D pattern first and then 2D information, because anthracene dimers would be de-cross-linked at elevated temperatures,⁴⁰ raising a risk of wiping the 2D fluorescent pattern if the sequence is reversed.” (Lines 351-354, Page 18, in the revised manuscript)

Fig. R1-7 The attempt for reversing the sequence of input of 3D/2D information; input the 2D information “0520” first and then input the 3D information “2022”.

References:

7. Van Damme, J. & Du Prez, F. Anthracene-containing polymers toward high-end applications. *Prog. Polym. Sci.* **82**, 92-119 (2018).
8. Tu, M., et al. Reversible optical writing and data storage in an anthracene-loaded metal-organic framework. *Angew. Chem. Int. Ed.* **131**, 2445-2449 (2019).
9. Van Damme, J., van den Berg, O., Vlamincx, L., Brancart, J., Van Assche, G. & Du Prez, F. Anthracene-based polyurethane networks: Tunable thermal degradation, photochemical cure and stress-relaxation. *Eur. Polym. J.* **105**, 412-420 (2018).
10. Yamamoto, T., Yagyu, S. & Tezuka, Y. Light-and heat-triggered reversible linear-cyclic topological conversion of telechelic polymers with anthryl end groups. *J. Am. Chem. Soc.* **138**, 3904-3911 (2016).

6. A minor comment: merging some of the supplementary movies into one is better.

Response: Thanks for the good comment. We have merged some of the supplementary movies into one as follows. The original Supplementary Movies 1-3 have been merged (Supplementary Movie 1), and the original Supplementary Movies 4-5 have been merged (Supplementary Movie 2).

Supplementary Movie 1-Demonstration of thermally-induced plasticity

Supplementary Movie 2-Recovery behaviors in successive shape-memory cycles

For Reviewer #2:

The manuscript entitled “Bioinspired Thermadapt Shape-Memory Polymer with Light-Induced Reversible Fluorescence for Rewritable 2D/3D-Encoding Information Carriers” described a thermally programmable shape-memory polymer materials with UV responsive fluorescence for data encryption and decryption. The shape memory effect was achieved through the thermal dynamic covalent bonds, while reversible fluorescent was achieved via anthracene motifs that undergoes photo-dimerization under UV irradiation. This polymer material shows double information encryption. The results are interesting. Revisions are required before the manuscript can be considered for publication.

Response: Thanks very much for reviewing our manuscript and giving us constructive comments. We have clarified the concerns in the following.

1) Generally, the innovative contributions of this paper required to be redefined and clarified throughout the paper. The material systems used in this manuscript, including shape memory effect of EVA and light-tunable fluorescence of anthracene motifs, have been studied in literatures. Introducing the concept of octopus biomimicry alone may risk exaggerating the function of the materials.

Response: Thanks for your good comment. In this study, we aimed to deliver the idea of integrating rewritable 2D/3D information encoding in one system to achieve high-security information carriers. In fact, this was inspired by the mimic octopus, which is capable of changing both color and shape to camouflage to escape from predators. We valued this biomimicry as an effective approach to realizing our 2D/3D-encoding

strategy. Then, our interest was focused on thermadapting SMPs and reversible photo-cross-linking, which were expected to account for the creation and encoding of 3D and 2D information, respectively. We would like to address the reviewer's concern in the following.

First, the design of anti-counterfeiting materials is always inspired by nature. At present, fluorescent materials are widely researched for anti-counterfeiting applications, and the octopus is the popular inspiration for such biomimicry. In fact, the octopus is a very popular inspiration for materials designs, including actuators, underwater robots, and dry/wet adhesives¹⁻⁶. For anti-counterfeiting materials, the color-shifting behavior of the octopus is quite attractive. For example, Ruijia Wang and coworkers recently designed an information encryption system for environmental interaction by mimicking the morphology/pattern information transfer function of luminescent octopus.⁷

Second, the concept of mimic octopus biomimicry inspires us to integrate 2D/3D information encoding in one system. Unlike most of the previous work focused on mimicking the color-changing behavior, we stumbled upon the mimic octopus and were impressed by its intelligence of changing both color and shape. We realized that the color-shifting should be 2D information encoding, while the shape-shifting can be a 3D one. Following this thinking and some of our previous research basis, we finally determined thermadapting shape-memory effect and reversible photo-cross-linking as the essential two properties of the target material.

Third, cEVA and anthracene are the two key components in the design of our TSFF, and their unique functions in our anti-counterfeiting system are carefully considered. As documented, cEVA holds excellent thermally shape-memory effect and two-way shape-memory effect, which has been used to design actuators and sensors. However, in this study, we were focused on the thermadapting shape-memory effect of cEVA, which was the core to rewritable 3D information encoding; here, the thermal adaptability of cEVA enabled the creation of 3D pattern (information input), while its shape-memory effect was responsible for the sheltering of the information (encoding). As these two processes are recoverable, the 3D information can be rewritable. To our knowledge, the design and use of cEVA in this work are rarely reported. Meanwhile, the reversible

photo-cross-linking of the anthracene group has been used to either tune the mechanical properties of a material or endow a material with light-induced shape-memory effect. In this study, we engineered the fluorescence-shifting behavior of anthracene as 2D encoding, and its light-induced reversibility made the 2D information rewritable. In all, the cEVA and anthracene chemistry were not simply merged, they are carefully considered and designed from the view of the integration of rewritable 2D/3D encoding. Anyhow, we appreciate the excellent reported works about anthracene and EVA as they facilitated our selection and design of proper materials.

Finally, we realized that the previous manuscript lacked a vivid display of the shape/color-shifting behavior of the mimic octopus, so we added a schematic diagram to present the biomimicry strategy in the revised manuscript, as shown in Fig. R2-1 (Supplementary Fig. 1 in the Supplementary Information).

We really appreciate this constructive comment from the reviewer.

Fig. R2-1 (Supplementary Fig. 1 in the Supplementary Information) Illustration of the biomimicry of the mimic octopus, which camouflages as a coral by shape/color-shifting.

References:

1. Zhao, Y., et al. Somatosensory actuator based on stretchable conductive photothermally responsive hydrogel. *Sci. Robot.* **6**, eabd5483 (2021).
2. Hwang, D., Barron, E.J., Haque, A. & Bartlett, MD. Shape morphing mechanical metamaterials through reversible plasticity. *Sci. Robot.* **7**, eabg2171 (2022).
3. Khodambashi, R., et al. Heterogeneous hydrogel structures with spatiotemporal reconfigurability using addressable and tunable voxels. *Adv. Mater.* **33**, 2005906 (2021).
4. Wu, Y., et al. Microtemplated electrowetting for fabrication of shape-controllable

- microdomes in extruded microsucker arrays toward octopus-inspired dry/wet adhesion. *Adv. Funct. Mater.* **33**, 2210562 (2023).
5. Wilker, JJN. How to suck like an octopus. *Nature* **546**, 358-359 (2017).
 6. Lee, Y.W., Chun, S., Son, D., Hu, X., Schneider, M. & Sitti, M. A. Tissue adhesion-controllable and biocompatible small-scale hydrogel adhesive robot. *Adv. Mater.* **34**, 2109325 (2022).
 7. Wang, R., et al. Bio-inspired structure-editing fluorescent hydrogel actuators for environment-interactive information encryption. *Angew. Chem. Int. Ed.* **62**, e202300417 (2023).

2) The reason for introducing the transesterification in EVA is not clear. If the objective is to introduce a permanent deformation, is it possible to design more complex multi-step deformations combining with crystallization/melting? Similarly, if the goal is to achieve reprocessability and self-welding, the author should provide more data on vitrimer properties rather than solely reporting the experimental results.

Response: Thanks for your excellent comment. First, in this work, the dynamic nature of cEVA (transesterification) was responsible for the creation of a certain 3D pattern (3D information input) and erasing it to make the information rewritable. Meanwhile, reprocessing and self-welding were added values for providing TSFF with sustainability and geometry diversity. Overall, the rewritable 3D information, reprocessing, and self-welding constituted the framework of the considerations on 3D encoding.

Second, designing more complex multi-step deformations via crystallization-melting transition of an SMP may be an approach to introduce permanent shape configuration, for instance, designing a thermo-responsive triple SMP, which usually consists of two segments with a distinct transition temperature (T_{trans}) and chemically cross-links; the polymer segments act as “molecular switches”, while the chemical cross-links act as “netpoints”. However, it is difficult to achieve reprocessing and self-welding properties, that is, the multi-functionality of the derived information carriers is limited. Although several works have reported the introduction of dynamic covalent bonds to endow such an SMP with reprocessing and self-welding property, its design

and synthesis is always complicated, especially compared with the commercial EVA, which simply involves the addition of catalyst during heating and thermal cross-linking. Following this idea, we designed and prepared TSFF for information carriers with rewritable 2D/3D encoding.

Third, as documented, properties of a shape-memory vitrimer usually include permanent shape reconfiguration, reprocessing, and self-welding.⁸⁻¹⁰ These properties rely on the dynamic nature of adaptive covalent bonds, for example, ester bonds, and the dynamic nature can be demonstrated in return. According to the reviewer's comment, we carried out additional experiments and discussions in the revised manuscript.

Permanent shape-reconfiguration. Commonly, cEVA cannot permanently change its original shape due to its insoluble and non-meltable characteristics. However, the dynamic nature of ester bonds would trigger the rearrangement of the polymer network, and a new permanent shape is obtained; this is the so-called thermally induced plasticity. Here, we demonstrated the permanent shape-reconfiguration of our TSFF by qualitative experiments and DMA tests, the results are now shown in Fig. R2-2 (Fig. 3d, e, f in the revised manuscript).

Fig. R2-2 (Fig. 3d, e, f in the revised manuscript) Thermadapt shape-memory property of TSFF. d Demonstration of the thermal plasticity of TSFF. e Demonstration of the thermal

plasticity and SME of TSFF. *f* Thermadapt SME of TSFF as measured by DMA. S_o , S_t , and S_r are defined as the original, temporary, and recovery shapes, respectively, and S_{np} is defined as the new permanent shape produced by thermally induced plasticity.

Reprocessing property. The dynamic nature of the ester bonds brings TSFF with reprocessing, just like vitrimers. In this work, we prepared new samples by hot pressing fragments, broken samples, or defective samples at 160 °C to achieve recycling, as shown in Fig. R2-3 (Fig. 4 a-d in the revised manuscript). Moreover, we examined the mechanical properties of the reprocessed samples by tensile tests. The results in Fig. R2-4 (Supplementary Fig. 10 in the Supplementary Information) show that Young’s modulus, tensile strength, and elongation at break are almost identical, indicating good reprocessing properties and sustainability of TSFF. To make the manuscript readable for broad readers, we have presented vivid pictures regarding reprocessing in the main text and test data in the Supplementary Information.

In comparison, we studied the reprocessing property of TSFF without adding TBD. The results in Fig. R2-5 (Supplementary Fig. 9 in the Supplementary Information) indicate that the pristine TSFF sample cannot be reprocessed. Therefore, it demonstrates the dynamic nature and thermally induced plasticity of TSFF in return. We have added the following text to the main text to clarify this comparison.

“On the contrary, the controlled sample without adding TBD catalyst cannot be reprocessed (Supplementary Fig. 9), and this result confirmed the dynamic nature of TSFF in return.” (Lines 257-259, Page 13, in the revised manuscript)

Fig. R2-3 (Fig. 4 a-d in the revised manuscript) a Illustration of reprocessing process. b

Molecular mechanism of reprocessability. **c** Demonstration of the reprocessing of a sample into a butterfly shape for the first time. **d** Demonstration of the reprocessing of a sample into a smiley-face shape for the second time.

Fig. R2-4 (Supplementary Fig. 10 in the Supplementary Information) Comparison of the mechanical properties between the original and reprocessed TSFF samples.

Fig. R2-5 (Supplementary Fig. 9 in the Supplementary Information) The controlled sample without TBD catalyst cannot be reprocessed.

Self-welding. We studied the welding property of TSFF by qualitative experiments and mechanical properties. The results in Fig. R2-6 (Fig. 4e-k in the revised manuscript) and Fig. R2-7 (Supplementary Fig. 14 in the Supplementary Information) demonstrate the excellent welding effect of TSFF as the welded sample shows a compatible interface topography and can lift a weight of 2 kg, 15000 times its own weight. Moreover, the results of tensile tests show that the welded TSFF does not break even after stretching to 1000%. In the revised manuscript, we have discussed the self-welding aspect of TSFF in Lines 268-299.

Fig. R2-6 (Fig. 4 e-k in the revised manuscript) Self-welding properties of TSFF. e Schematic illustration of self-welding process. **f** Photographic image showing typical self-welding behavior: First, the dog bone-shaped sample is cut into two halves by a blade and then welded at 160 °C for 4 h. **g** Scanning electron microscopy (SEM) image of a TSFF sample after welding. **h** Visual demonstration of stretching different strains (0%, 200%, 400%, 600%, 800%, and 1000%) after 4-h welding of TSFF samples. **i** Stress-strain curves of TSFF samples at different times of welding (0.5, 1, 2, and 4 h). **j** Schematic illustration of assembling a 3D flower by welding 2D petals layer-by-layer. **k** Visual demonstration of thermally induced SME and fluorescence of welded 3D flowers.

Fig. R2-7 (Supplementary Fig. 14 in the Supplementary Information) Photograph of a self-welded sample that can hold a 2 kg weight.

References:

- Zheng, N., Xu, Y., Zhao, Q. & Xie, T. Dynamic covalent polymer networks: a molecular platform for designing functions beyond chemical recycling and self-healing. *Chem. Rev.* **121**, 1716-1745 (2021).

9. Yang, Y., Xu, Y., Ji, Y. & Wei, Y. Functional epoxy vitrimers and composites. *Prog. Mater. Sci.* **120**, 100710 (2021).
10. Luo, J., et al. Elastic vitrimers: Beyond thermoplastic and thermoset elastomers. *Matter* **5**, 1391-1422 (2022).

3) Anthracene was employed as a tunable fluorescent moiety, which was often used for modulating mechanical properties and achieving shape memory effects. The authors need to provide further discussion regarding the influence of the photo-dimerization of anthracene on the mechanical properties and shape memory effect. Also, does the formation of covalent bonds through photo-dimerization decrease the reprocessability of the material?

Response: Thank you for your comment. In this work, anthracene was employed as the fluorescence-shifting component to realize 2D information input and encoding. As well, we agree that anthracene is also often used to either modulate mechanical properties or achieve shape memory effects, as reported by a lot of literature, including our previous works.¹¹⁻¹⁵ Our response to this comment is in the following.

First, photo-cross-linking had no significant effect on the mechanical properties of TSFF. Indeed, several literatures have reported the use of anthracene chemistry to modulate the mechanical properties of a material.¹¹⁻¹³ In our current work, the content of anthracene in the material system is very small, and it is only responsible for providing the functions of fluorescence and fluorescence-shifting. Thus, the photo-cross-linking of anthracene has an ignorable effect on the mechanical properties of TSFF theoretically. Then, we carried out additional experiments to verify this speculation. In Fig. R2-8, the tensile strength, Young's modulus, and elongation at break are almost identical for the samples with and without 365 nm UV irradiation. This result demonstrates our conclusion.

Fig. R2-8 **a** Stress-strain curves of the original and UV-irradiated samples. **b** Comparison of mechanical properties of the original and UV-irradiated samples.

Second, photo-cross-linking of the anthracene groups had no significant effect on the shape-memory effect of TSFF. In this study, the thermadap shape-memory effect of TSFF was merely enabled by the cEVA component, the reversible photo-cross-linking of anthracene was not involved. As documented, anthracene has been used to design light-induced SMPs where both the shape programming and recovery depend on the reversible photo-cross-links.¹⁴⁻¹⁵ However, in the current work, the shape memory effect of TSFF was dominated by the crystallization-melting transition of the cEVA, the photo-dimerization of the small amount of anthracene moieties should not be responsible for that. Meanwhile, we carried out additional experiments to examine the shape-memory properties of samples with and without UV irradiation. The results demonstrate that the photo-cross-linking of anthracene has no significant effect on shape-memory properties as both the original and UV-irradiated TSFF samples hold excellent shape-memory effect (Fig. R2-9).

Fig. R2-9 Comparison of the shape memory properties of the original sample and UV-irradiated sample.

Finally, the results of our additional experiments indicate that photo-cross-linking has no significant effect on the reprocessing property of TSFF. Although the reversible efficiency of anthracene dimers showed a gradual decrease trend in successive cycles, the reprocessing property mostly relied on the dynamic exchange of ester bonds. Besides, the small amount of PCL segment is also dynamic because of the ester bonds.¹⁶⁻²⁰ Therefore, the reprocessing property of TSFF should not be obviously affected by anthracene. Moreover, we did additional experiments to verify this conclusion; the UV-irradiated TSFF sample still hold excellent reprocessing property (Fig. R2-10), directly supporting the conclusion.

Fig. R2-10 Reprocessing property of the UV-irradiated sample.

References:

11. Van Damme, J., van den Berg, O., Vlamincx, L., Brancart, J., Van Assche, G. & Du Prez, F. Anthracene-based polyurethane networks: Tunable thermal degradation, photochemical cure and stress-relaxation. *Eur. Polym. J.* **105**, 412-420 (2018).
12. Huq, N.A., Ekblad, J.R., Leonard, A.T., Scalfani, V.F. & Bailey, T.S. Phototunable thermoplastic elastomer hydrogel networks. *Macromolecules* **50**, 1331-1341 (2017).
13. Van Damme, J. & Du Prez, F. Anthracene-containing polymers toward high-end applications. *Prog. Polym. Sci.* **82**, 92-119 (2018).
14. Xie, H., Yang, K.-K. & Wang, Y.-Z.. Photo-cross-linking: a powerful and versatile strategy to develop shape-memory polymers. *Prog. Polym. Sci.* **95**, 32-64 (2019).
15. Xie, H., et al. Design of poly (L-lactide)-poly (ethylene glycol) copolymer with light-induced shape-memory effect triggered by pendant anthracene groups. *ACS Appl. Mater. Interfaces* **8**, 9431-9439 (2016).
16. Zhao, Q., Zou, W., Luo, Y. & Xie, T. Shape memory polymer network with thermally distinct elasticity and plasticity. *Sci. Adv.* **2**, e1501297 (2016).
17. Du, L., Dai, J., Xu, Z.-Y., Yang, K.-K. & Wang, Y.-Z. From shape and color memory PCL network to access high security anti-counterfeit material. *Polymer* **172**, 52-57 (2019).
18. Chen, G., et al. Converse two-way shape memory effect through a dynamic covalent network design. *J. Mater. Chem. A* **10**, 10350-10354 (2022).
19. Miao, W., Zou, W., Luo, Y., Zheng, N., Zhao, Q., Xie, T. Structural tuning of polycaprolactone based thermadap shape memory polymer. *Polym. Chem.* **11**, 1369-1374 (2020).
20. Song, H., Fang, Z., Jin, B., Pan, P., Zhao, Q. & Xie, T. Synergetic chemical and physical programming for reversible shape memory effect in a dynamic covalent network with two crystalline phases. *ACS Macro Lett.* **8**, 682-686 (2019).

4) The disassociation of anthracene dimers can also be thermally induced. When programming the shape by heating, did the fluorescence pattern get affected?

Response: We really appreciate the reviewer's comment as it helps us to better clarify our 2D/3D encoding system. First, we agree with the reviewer's point that the

disassociation of anthracene dimers can be thermally triggered at elevated temperatures.²¹⁻²⁴ Thus, there is a risk of wiping the fluorescence patterns during thermally-induced plasticity at 120 °C if the sequence of 2D and 3D patterns (information) had not been carefully considered. If we create a 2D fluorescence pattern by 365 nm UV irradiation first and then create a 3D pattern by thermally induced plasticity at 120 °C, the anthracene dimers seem to be disassociated, and the 2D pattern becomes unclear (Fig. R2-11). To avoid this impact, we have carefully considered the input sequence of 2D/3D information, that is, creating a 3D surface pattern first, followed by the creation of a 2D fluorescent pattern. In this way, the effect of heating on anthracene dimers could be eliminated. As for the macroscopically programming at 75 °C, our results of cyclic 2D/3D encoding (Fig. 6) indicated that there was no significant impact on the 2D fluorescent patterns as it did not reach the disassociation temperature of anthracene dimers.

Fig. R2-11 The attempt for reversing the sequence of input of 3D/2D information; input the 2D information “0520” first and then input the 3D information “2022”.

References:

21. Van Damme, J. & Du Prez, F. Anthracene-containing polymers toward high-end applications. *Prog. Polym. Sci.* **82**, 92-119 (2018).
22. Tu, M., et al. Reversible optical writing and data storage in an anthracene-loaded metal-organic framework. *Angew. Chem. Int. Ed.* **131**, 2445-2449 (2019).
23. Van Damme, J., van den Berg, O., Vlamincx, L., Brancart, J., Van Assche, G. & Du Prez, F. Anthracene-based polyurethane networks: Tunable thermal degradation, photochemical cure and stress-relaxation. *Eur. Polym. J.* **105**, 412-420 (2018).
24. Yamamoto, T., Yagyu, S. & Tezuka, Y. Light-and heat-triggered reversible linear-cyclic

topological conversion of telechelic polymers with anthryl end groups. *J. Am. Chem. Soc.* **138**, 3904-3911 (2016).

5) In Figure 4i, why the welded specimen shows higher stiffness than the pristine one?

Response: Thanks for your good comment. The results of mechanical properties of the welded TSFF samples indicate that the welded specimen holds higher tensile stress than the original one, especially the samples that were welded for 2 and 4 h (Fig. 4i). This phenomenon could be explained by the fact that the welded samples had a higher cross-section area, improving the resistance of the welded specimen to external deformation. As shown in Fig. 4h, the welded part shows a relatively smaller strain increment than the unwelded parts when stretching to 1000%. Besides, a longer time of hot-pressing could contribute to a more dense and compatible welding interface (Fig. 4g), which can also improve the tensile stress. Fig. 4i further indicates that the welding efficiency depends on the welding time because tensile stress increases when welding time increases.

Fig. R2-12 (Fig. 4 e-i in the revised manuscript) Reprocessing and self-welding properties of TSFF. e Schematic illustration of self-welding process. f Photographic image showing typical self-welding behavior: First, the dog bone-shaped sample is cut into two halves by a blade and then welded at 160 °C for 4 h. g Scanning electron microscopy (SEM) image of a TSFF sample after welding. h Visual demonstration of stretching different strains (0%, 200%, 400%, 600%, 800%, and 1000%) after 4-h welding of TSFF samples. i Stress-strain curves of TSFF samples at different times of welding (0.5, 1, 2, and 4 h).

For Reviewer #3:

This work reports a kind of functional polymers with thermadappt shape-memory effect and reversible fluorescence-shifting properties for high-security information carriers. The idea that integrates 2D and 3D encoding in a single carrier is interesting, and this is significant for improving information security. The authors also confirm that the reversible photo-cross-linking and thermally-induced plasticity can make both 2D and 3D information rewritable, this endows the derived information carriers with reusability. The experiments, results, and discussions presented in the manuscript are clear. In all, this work will inspire the development of multifunctional and high-security information carriers, I recommend it for publication in Nature Communications. However, the authors should clarify the following issues clearly.

Response: Thanks for your recommendation. We have made changes in the manuscript accordingly. The following is our point-by-point response to the comments.

1. In the Introduction, the authors claimed that a hydrogel-based information carrier is susceptible to dehydration, what the situation would be if the hydrogel has been designed to be anti-dehydration? Meanwhile, what about the long-term stability of the TSFF in the current work?

Response: Thanks for your good comment. First, hydrogels contain abundant water, and the evaporation of water is spontaneous; this is the dehydration of hydrogels. The gradual dehydration of a hydrogel would result in the weakness of flexibility and even make itself brittle. Several methods have been proposed to equip hydrogels with anti-dehydration capacity, for example, introducing highly hydrating salts or alcohols, but the dehydration process can only be retarded, not fully stopped.¹ This means that the hydrogels would still lose water gradually. Therefore, we thought that a hydrogel-based information carrier might be still faced with the susceptibility to dehydration even if it has been designed to be anti-dehydration. Anyhow, we still recognized that hydrogels are available materials for information carriers not requiring long-stability.

Second, we have examined the long-term stability of our TSFF as recommended by the reviewer, and now we would like to clarify the results of the long-term stability

of TSFF in the following points. The first thing is that TSFF has good long-term stability in a natural environment. As shown in Fig. R3-1 (Supplementary Fig. 6 in the Supplementary Information), we stored the patterned TSFF in the natural environment for up to 8 months, the material and its fluorescent pattern showed no obvious change. This result confirms the long-stability of 2D fluorescent information. Meanwhile, TSFF has good stability in chemicals. As shown in Fig. R3-2 (Supplementary Fig. 7 in the Supplementary Information), the patterned TSFF showed no obvious change when immersing in strong acid (HCl), strong alkaline (NaOH), and salt solution (NaCl) for 240 days. These results demonstrate the feasibility of TSFF as long-stable information carriers.

Fig. R3-1 (Supplementary Fig. 6 in the Supplementary Information) Demonstration of long-term stability by storing a patterned TSFF sample in natural environment for 8 months.

Fig. R3-2 (Supplementary Fig. 7 in the Supplementary Information) Demonstration of chemical stability of the patterned TSFF by immersing the samples in strong acidic (HCl),

strong base (NaOH), and salt solution (NaCl) at ambient temperature for 240 days, respectively.

Reference:

1. Wu, S., Guo, J., Wang, Y., Xie, H. & Zhou, S. Cryopolymerized polyampholyte gel with antidehydration, self-healing, and shape-memory properties for sustainable and tunable sensing electronics. *ACS Appl. Mater. Interfaces* **14**, 42317-42327 (2022).

2. The authors stated that the shape memory effect in the previous works was merely used for information encasement and not for practical information (Line 60), so this work proposes TSFF with a thermadapt shape memory effect. The authors did not clearly state why the thermadapt shape memory effect can address these concerns. In addition, the authors should keep the consistency of expression with respect to “thermadapt” in each section.

Response: Thanks for your good comment. In this work, the thermadapt shape-memory effect holds two core functions, the creation of a 3D pattern (information) by thermally induced plasticity and the encasement of the 3D information. Most of the previous works were focused on the latter aspect. Here, we clarify our idea regarding the thermadapt shape-memory effect as follows.

First, the thermadapt shape-memory effect would facilitate the creation of rewritable 3D patterns via thermally induced plasticity, that is, the reconfiguration of the permanent shape of an SMP. Just as we described in the main text, the commercial technique for creating 3D information uses certain templates and 3D printing. However, these 3D patterns cannot be reprogrammed, making the 3D information not erasable or rewritable; this is not beneficial to a sustainable material platform. Based on the advantages of the shape-shifting feature of SMPs, researchers have also used templates when deforming SMPs and constructing certain 3D patterns. However, one contradiction is that a 3D pattern created in this way will be wiped when using the SME to encase it; otherwise, it cannot be encrypted. Therefore, SMPs capable of reconfiguring permanent 3D patterns are in high demand. Thus, we proposed thermadapt SMPs as the promising material platform that enables the facile fabrication

of rewritable 3D patterns.

Second, the shape-memory effect following the creation of rewritable 3D patterns would enable the encoding of 3D information by macroscopically shape programming. Just as we stated in the main text, to encrypt the former 3D information using the SME, a flat TSFF is programmed into a folded airplane-like shape, so the 3D information has been sheltered inside and cannot be detected. After the multi-stage 2D/3D encoding, all of the 2D and 3D information can be identified by designated inspectors by heating the TSFF for shape recovery and exposing it to UV light in sequence.

In all, the relevant descriptions above have been modified in the revised text for a clearer discussion.

Finally, we would like to introduce the origin of thermadapt SMPs to address the reviewer's concern about consistency. As reported, Prof. Tao Xie's group proposed the concept of thermadapt SMPs. In detail, thermadapt SMPs refer to SMP networks with dynamic covalent connections that reconfigure their permanent shapes through solid-state plasticity, which is different from the traditional categories of thermoplastic SMPs and thermoset SMPs. They suggested using the term "thermadapt SMP" to define the SMP networks with unique thermally elasticity and plasticity. This is because it not only resonates with thermoplastic and thermoset SMPs but also describes the unique behavior, i.e., thermal adaptability.² In this work, as the cEVA component possessed thermal adaptability, we used the item, "thermadapt". Anyhow, to avoid confusion, we use "thermally induced plasticity" to describe reprocessing, self-welding, and the creation of 3D patterns (information) in the revised manuscript.

Reference:

2. Zou, W., Dong, J., Luo, Y., Zhao, Q. & Xie, T. Dynamic covalent polymer networks: from old chemistry to modern day innovations. *Adv. Mater.* **29**, 1606100 (2017).

3. The TSFF in this work holds the property of self-welding, as demonstrated in Figure 4, but the authors rarely discuss its connection with information storage, the core of this work. What benefits can the self-welding property bring to information anti-

counterfeiting?

Response: Thanks for your good comment. In the current work, we considered the abundance of dynamic ester bonds in the TSFF, which could feasibly bring reprocessing and self-welding properties. From the view of an information carrier, reprocessing and self-welding properties may provide it with sustainability and information diversity.

First, the self-welding property of TSFF enables the assembly of complex geometries, similar to the traditional Chinese mortise and tenon technologies for interconnecting simple components for wooden buildings. This enables the creation and encoding of a 2D fluorescent pattern in a complex 3D geometry. As shown in Fig. R3-3 (Supplementary Fig. 15 in the Supplementary Information), owing to the reversible fluorescence-shifting of the TSFF, it can be used to create a 2D pattern “W” on a petal of the flower-like structure generated from self-welding. The natural light-invisible 2D pattern can be encrypted after folding.

Second, the self-welding property of TSFF may enable the creation of 3D patterns directly onto the TSFF. Here, we carried out additional experiments to demonstrate this concept to the reviewers and readers. As shown in Fig. R3-4 (Supplementary Fig. 16 in the Supplementary Information), we have successfully welded three “SOS” TSFF samples with a piece of TSFF, which can be regarded as a certain 3D information. We think it may be an available strategy for 3D information input besides thermally induced plasticity.

Fig. R3-3 (Supplementary Fig. 15 in the Supplementary Information) Photograph of the flower produced by welding several petal-like TSFF samples and its shape-memory properties. The 2D information “W” was printed in the center.

Fig. R3-4 (Supplementary Fig. 16 in the Supplementary Information) The proof-of-concept of using self-welding property to create 3D information.

4. The pictures in Figure 4e showing the 2D information encoding of TSFF are not well matched, the areas (highlighted by red frames) camouflaged by several TSFF specimens are not consistent with those after exposure to UV light. For instance, the authors can compare the word “BANK” before and after UV exposure. The authors should carefully mark them to avoid any confusion.

Response: Thanks for your comment. We realized this carelessness occurred in Fig. 5e, and we have revised it accordingly.

Fig. R3-5 (Fig. 5 in the revised manuscript) 2D-encoding anti-counterfeiting strategy. **a** Two pieces of TSFF are adhered to a gift box as labels. **b** QR code visible at UV light (254 nm) and decoded information. **c** A few sentences visible under UV light (254 nm). **d** TSFF label

attached to a glass bottle. The fluorescent pattern is the Sun and Immortal Birds (the logo of the city of Chengdu). e Integration of multiple encrypted information into banknote model on TSFF. Top left: Letter RON; Right: Panda pattern; Bottom left: ¥100; Middle: Barcode pattern.

5. As documented, photo-patterning is also a conventional approach to generate 3D geometries from a flat 2D film upon localized light irradiation, is there any possibility that the TSFF became 3D when precisely creating 2D surface patterns?

Response: Thanks for your good comment. Indeed, photo-patterning is a good strategy to produce 3D geometries from a 2D flat film. To our knowledge, it always requires external stress (e.g., solid films) or a gradient structure (e.g., hydrogels).³⁻⁸ In this work, the TSFF bears no external stress when irradiated by a 365 nm UV light, that is, the creation of a 2D fluorescent pattern, so the TSFF is unlikely to become 3D during this process. Besides, the content of the anthracene moieties is relatively small. We have carried out experiments to verify this speculation directly. As shown in Fig. R3-6 (Supplementary Fig. 5 in the Supplementary Information), we did not observe the 2D to 3D shape transformation of TSFF in three successive photo-patterning cycles.

Fig. R3-6 (Supplementary Fig. 5 in the Supplementary Information) The cyclic process of fluorescent patterns of Bing Dwen Dwen, Rongbao and Shuey Rhon Rhon.

References:

3. Qiu, Y., Munna, D.-R., Wang, F., Xi, J., Wang, Z. & Wu, D. Regulating asynchronous

- deformations of biopolyester elastomers via photoprogramming and strain-induced crystallization. *Macromolecules* **54**, 5694-5704 (2021).
4. Zhu, C.N., et al. Reconstructable gradient structures and reprogrammable 3D deformations of hydrogels with coumarin units as the photolabile crosslinks. *Adv. Mater.* **33**, 2008057 (2021).
 5. Wang, Y., et al. Light-activated shape morphing and light-tracking materials using biopolymer-based programmable photonic nanostructures. *Nat. Commun.* **12**, 1651 (2021).
 6. Bai, J. & Shi, Z. Shape memory: an efficient method to develop the latent photopatterned morphology for elastomer in two/three dimension. *ACS Macro Lett.* **6**, 1025-1030 (2017).
 7. Bai, J., Shi, Z., Yin, J., Tian, M. & Qu, R. Shape reconfiguration of a biomimetic elastic membrane with a switchable Janus structure. *Adv. Funct. Mater.* **28**, 1800939 (2018).
 8. Jiang, Z., et al. Strong, self-healable, and recyclable visible-light-responsive hydrogel actuators. *Angew. Chem. Int. Ed.* **132**, 7115-7122 (2020).

6. The synthesis route of AC-PCL-AN presented in Supplementary Fig. 1 is not consistent with the details described in the Experimental section, and the chemical structure of AC-PCL-AN in Figure 2 should be checked accordingly.

Response: Thanks for your comment. We noticed that the synthesis route of AC-PCL-AN appeared inaccuracy; the temperature and time for the reaction should be 140 °C and 12 h, respectively, as updated in Supplementary Fig. 2b (revised version). In addition, we revised an error in the chemical structure of AC-PCL-AN in its ¹H-NMR spectrum, as updated in Supplementary Fig. 3 (revised version).

Fig. R3-7 (Supplementary Fig. 2 in the Supplementary Information) Synthesis route of

photoactive material AC-PCL-AN.

Fig. R3-8 (Supplementary Fig. 3 in the Supplementary Information) ¹H-NMR characterization of AC-PCL-AN.

REVIEWERS' COMMENTS

Reviewer #1 (Remarks to the Author):

The manuscript has been carefully revised based on the reviewers' comments. The required data is also provided. The current version is acceptable for publication.

Reviewer #2 (Remarks to the Author):

The authors have addressed carefully the issues raised by the reviewers in the response letter and in the manuscript. Additional results are added that have improved the quality of this work. Now it is suitable for publication. An acceptance is thereby recommended.

Reviewer #3 (Remarks to the Author):

The manuscript have been revised carefully according to the reviewers and can be accepted for publishing.